# Land use change and El Niño-Southern Oscillation drive decadal carbon balance shifts in Southeast Asia

Masayuki Kondo [1,2], Kazuhito Ichii [1,2,3], Prabir K. Patra [2,22], Joseph G. Canadell [4], Benjamin Poulter [5,6], Stephen Sitch[7], Leonardo Calle [5], Yi Y. Liu [8,9], Albert I.J.M. van Dijk[10], Tazu Saeki[3], Nobuko Saigusa[3], Pierre Friedlingstein [7], Almut Arneth[11], Anna Harper [7], Atul K. Jain[12], Etsushi Kato [13], Charles Koven [14], Fang Li [15], Thomas A.M. Pugh[11,16], Sönke Zaehle[17], Andy Wiltshire[18], Frederic Chevallier[19], Takashi Maki[20], Takashi Nakamura[21], Yosuke Niwa[20] & Christian Rödenbeck[17]

An integrated understanding of the biogeochemical consequences of climate extremes and land use changes is needed to constrain land-surface feedbacks to atmospheric $CO_2$ from associated climate change. Past assessments of the global carbon balance have shown particularly high uncertainty in Southeast Asia. Here, we use a combination of model ensembles to show that intensified land use change made Southeast Asia a strong source of $CO_2$ from the 1980s to 1990s, whereas the region was close to carbon neutral in the 2000s due to an enhanced $CO_2$ fertilization effect and absence of moderate-to-strong El Niño events. Our findings suggest that despite ongoing deforestation, $CO_2$ emissions were substantially decreased during the 2000s, largely owing to milder climate that restores photosynthetic capacity and suppresses peat and deforestation fire emissions. The occurrence of strong El Niño events after 2009 suggests that the region has returned to conditions of increased vulnerability of carbon stocks.

[1] Center for Environmental Remote Sensing (CEReS), Chiba University, Chiba 263-8522, Japan. [2] Department of Environmental Geochemical Cycle Research, Japan Agency for Marine-Earth Science and Technology, Yokohama 236-0001, Japan. [3] Center for Global Environmental Research, National Institute for Environmental Studies, Tsukuba 305-8506, Japan. [4] Global Carbon Project, CSIRO Oceans and Atmosphere, Canberra, ACT 2601, Australia. [5] Institute on Ecosystems and Department of Ecology, Montana State University, Bozeman, MT 59717, USA. [6] Biospheric Science Laboratory, NASA Goddard Space Flight Center, Greenbelt, MD 20771, USA. [7] University of Exeter, Exeter EX4 4QF, UK. [8] School of Geography and Remote Sensing, Nanjing University of Information Science and Technology, Nanjing 210044, China. [9] ARC Centre of Excellence for Climate Systems Science and Climate Change Research Centre, University of New South Wales, Sydney, NSW 2052, Australia. [10] Fenner School of Environment and Society, Australian National University, Canberra, ACT 0200, Australia. [11] Institute of Meteorology and Climate Research, Environmental Atmospheric Research (IMK-IFU), Karlsruhe Institute of Technology (KIT), Kreuzeckbahnstraße 19, 82467 Garmisch-Partenkirchen, Germany. [12] Department of Atmospheric Sciences, University of Illinois at Urbana-Champaign, Urbana, IL 61801, USA. [13] Institute of Applied Energy, Tokyo 105-0003, Japan. [14] Earth Sciences Division, Lawrence Berkeley National Laboratory, Berkeley, CA 94720, USA. [15] International Center for Climate and Environmental Sciences, Institute of Atmospheric Physics, Chinese Academy of Sciences, Beijing 100864, China. [16] School of Geography, Earth and Environmental Science and Birmingham Institute of Forest Research, University of Birmingham, Birmingham B15 2TT, UK. [17] Biogeochemical Integration Department, Max Planck Institute for Biogeochemistry, 07701 Jena, Germany. [18] Met Office Hadley Centre, Fitzroy Road, Exeter EX1 3PB, UK. [19] Laboratoire des Sciences du Climat et de l'Environnement (LSCE), CEA CNRS UVSQ, 91191 Gif Sur Yvette, France. [20] Meteorological Research Institute, Tsukuba 305-0052, Japan. [21] Japan Meteorological Agency, Tokyo 100-8122, Japan. [22] Present address: Research and Development Center for Global Change (RCGC) Japan Agency for Marine-Earth Science and Technology, Yokohama 236-0001, Japan. These authors contributed equally: Kazuhito Ichii, Prabir K. Patra, Joseph G. Canadell, Benjamin Poulter, Stephen Sitch, Leonardo Calle.  Correspondence and requests for materials should be addressed to M.K. (email: redmk92@gmail.com)

Southeast Asia is unique among tropical regions because the region is highly susceptible to the influence of El Niño-Southern Oscillation (ENSO)[1, 2] and is subject to highest deforestation rates in the tropical regions[3–6]. In the recent past, Southeast Asia experienced large $CO_2$ emissions, ranging from 0.81 to 1.2 Pg C yr$^{-1}$ due to the drought-induced fires during the 1997/1998 El Niño[7, 8], and a substantial loss of forest area (2.5 Mha in the 1990s)[9] due to forest conversion to oil palm and rubber tree plantations[10–12]. However, contrary to the intensively studied Amazon Basin and Congo Basin with permanent plot sample data[13, 14], the recent states of net $CO_2$ flux (balance between $CO_2$ uptake and release by the land biosphere) across Southeast Asia remain highly uncertain. The Fifth Assessment Report (AR5) of the Intergovernmental Panel on Climate Change (IPCC) provided a synthesis of the global net $CO_2$ flux. However, disregard of land use change (LUC) in the biosphere models (the bottom-up approach) resulted in an underestimation of $CO_2$ release for the tropical regions when compared with the atmospheric $CO_2$ inversions (the top-down approach)[15]. Both climate and LUC need to be integrated into analyses to adequately estimate the net $CO_2$ flux and to reconcile results from different approaches. Such an integrated effort has not been undertaken for Southeast Asia.

Here we investigate the decadal variability of the net $CO_2$ flux (termed Net Biome Production: NBP, the negative sign (−) for a net sink and the positive sign (+) for a net source) in Southeast Asia over the period 1980–2009 using an ensemble of seven terrestrial biosphere model simulations from the TRENDY model intercomparison project (Supplementary Table 1), an ensemble of five atmospheric $CO_2$ inversions that cover longer than two decades (Supplementary Tables 2, 3) and a remote-sensing-based annual biomass change estimated by Global Aboveground Biomass Carbon version 1.0 (Supplementary Fig. 1). We demonstrate that consideration of LUC processes to biosphere models brings consistency in interannual and decadal variability of the net $CO_2$ flux between the bottom-up, top-down and remote-sensing-based approaches, indicating carbon balance shifts towards a net source from the 1980s to 1990s, and towards a net sink from the 1990s to 2000s. Subsequently, we quantify the contributions to the decadal NBP variability from $CO_2$ fertilization, climatic conditions and LUC using three sets of TRENDY simulations (Supplementary Fig. 2) where biosphere models were forced with varying $CO_2$, climate and historical LUC (TRENDY S3), along with simulations forced with varying $CO_2$ and climate (TRENDY S2), and varying $CO_2$ only (TRENDY S1). Our results show that increased LUC emissions during the 1990s was the major factor responsible for the shift towards a net source between the 1980s and 1990s, and the enhanced $CO_2$ fertilization and absence of strong El Niño events during the 2000s for the shift towards a net sink between the 1990s and 2000s. The milder climate sustained during the 2000s is of particular importance to a high carbon assimilation by plant ecosystems in Southeast Asia, inducing a strong net uptake that cancels a large proportion of $CO_2$ release from ongoing LUC in the region.

## Results

### The effect of LUC on net $CO_2$ flux.
We find agreement in interannual variability of NBP between the TRENDY S3 and atmospheric $CO_2$ inversions for the period 1980–2009, and the annual biomass change (hereafter, Δbiomass) for the period 1994–2009, as indicated by high correlations between the three estimates ($r = 0.67$–$0.70$, $p < 0.01$, Fig. 1a; detailed inter-model comparisons in Supplementary Figs. 3, 4). However, this agreement is not found in a comparison with the TRENDY S2, which indicates continuously strong $CO_2$ uptake throughout the 30-year

period. The inclusion of LUC (adding LUC to the model forcing as in TRENDY S3; Supplementary Fig. 5) changed both the patterns of the spatial variability of NBP (Fig. 1b; individual model results in Supplementary Fig. 6) as well as the sign of mean annual NBP from a large sink to a weak source for the period 1980–2009; flux changed from −0.18 ± 0.09 Pg C yr$^{-1}$ in the TRENDY S2 (average ± 1$\sigma$ as model-by-model variability) to 0.09 ± 0.12 Pg C yr$^{-1}$ in the TRENDY S3. This result confirms that the LUC emissions are a key factor in NBP estimation for Southeast Asia, as its contribution to NBP is large, so much as to cancel the $CO_2$ uptake due to the effect of $CO_2$ fertilization.

### The decadal shifts of net $CO_2$ flux and attributions.
The inter-decadal mean NBP estimates from the TRENDY S3, atmospheric $CO_2$ inversions and Δbiomass yield a consistent pattern of decadal variability, indicating that an increased net source from the 1980s to the 1990s is largely decreased in the 2000s (Fig. 1c; individual model results in Supplementary Fig. 7). By isolating the contributions of the effects from $CO_2$ fertilization, climate and LUC to NBP using the TRENDY model simulations (Methods), we found that the shift towards a stronger net source from the 1980s to the 1990s is primarily attributable to the intensifying LUC (Fig. 1d; individual model results in Supplementary Fig. 8a), which increased the net source from 0.21 ± 0.11 Pg C yr$^{-1}$ (the 1980s) to 0.31 ± 0.13 Pg C yr$^{-1}$ (the 1990s). In contrast, a reduced net source (i.e. stronger sink) from the 1990s to the 2000s is attributed to the effects of $CO_2$ fertilization and climate (Fig. 1d; Supplementary Fig. 8b, c), with the former inducing a change from −0.23 ± 0.08 Pg C yr$^{-1}$ (the 1990s) to −0.28 ± 0.09 Pg C yr$^{-1}$ (the 2000s), and the latter from 0.09 ± 0.11 Pg C yr$^{-1}$ (the 1990s) to 0.001 ± 0.11 Pg C yr$^{-1}$ (the 2000s).

We examine the robustness of these decadal shifts by applying non-parametric trend tests to the periods 1980–1999 and 1990–2009 (Mann–Kendall and Theil slope tests; Methods). Using the NBP estimates including individual TRENDY models, we found that trends of increasing $CO_2$ uptake for the period 1990–2009 tend to be more statistically significant ($p < 0.05$) than those of increasing $CO_2$ release for the period 1980–1999 (Fig. 2a). Further analysis of the individual TRENDY models shows statistically significant trends of increasing $CO_2$ release due to the LUC effect for the period 1980–1999 and of increasing $CO_2$ uptake in response to the $CO_2$ fertilization and climate effects for the period 1990–2009 (Fig. 2b, c). This multi-model trend analysis suggests that the decadal NBP shift from the 1990s to 2000s is more robust than that from the 1980s to 1999s, which means that the $CO_2$ fertilization and climate conditions in the 2000s are more influential to NBP than the enhanced LUC activities in the 1990s, distinguishing the 2000s from previous decades. It is reasonable to expect that the $CO_2$ fertilization is partly responsible for the decadal NBP shift from the 1990s to 2000s because atmospheric $CO_2$ is the main factor driving NBP towards a net sink via promoting the photosynthetic carbon fixation[16]. The increased $CO_2$ fertilization effect in the 2000s coincides with higher $CO_2$ concentrations in the 2000s (379 ppm) than those in the 1980s (346 ppm) and the 1990s (361 ppm, decadal averages based on flask sampling data at the Mauna Loa Observatory: Data—NOAA Earth System Research Laboratory, https://www.esrl.noaa.gov/gmd/ccgg/trends/data.html). As for the climate effect, a net source in the 1980s and 1990s makes a transition to carbon neutral in the 2000s (Fig. 1d), implying a climate favourable for $CO_2$ uptake in the 2000s.

### Weak phase of ENSO in the 2000s.
In order to elucidate the climate effect on NBP in the 2000s, we have analysed mean annual NBP (i.e. the TRENDY S3, atmospheric $CO_2$ inversion

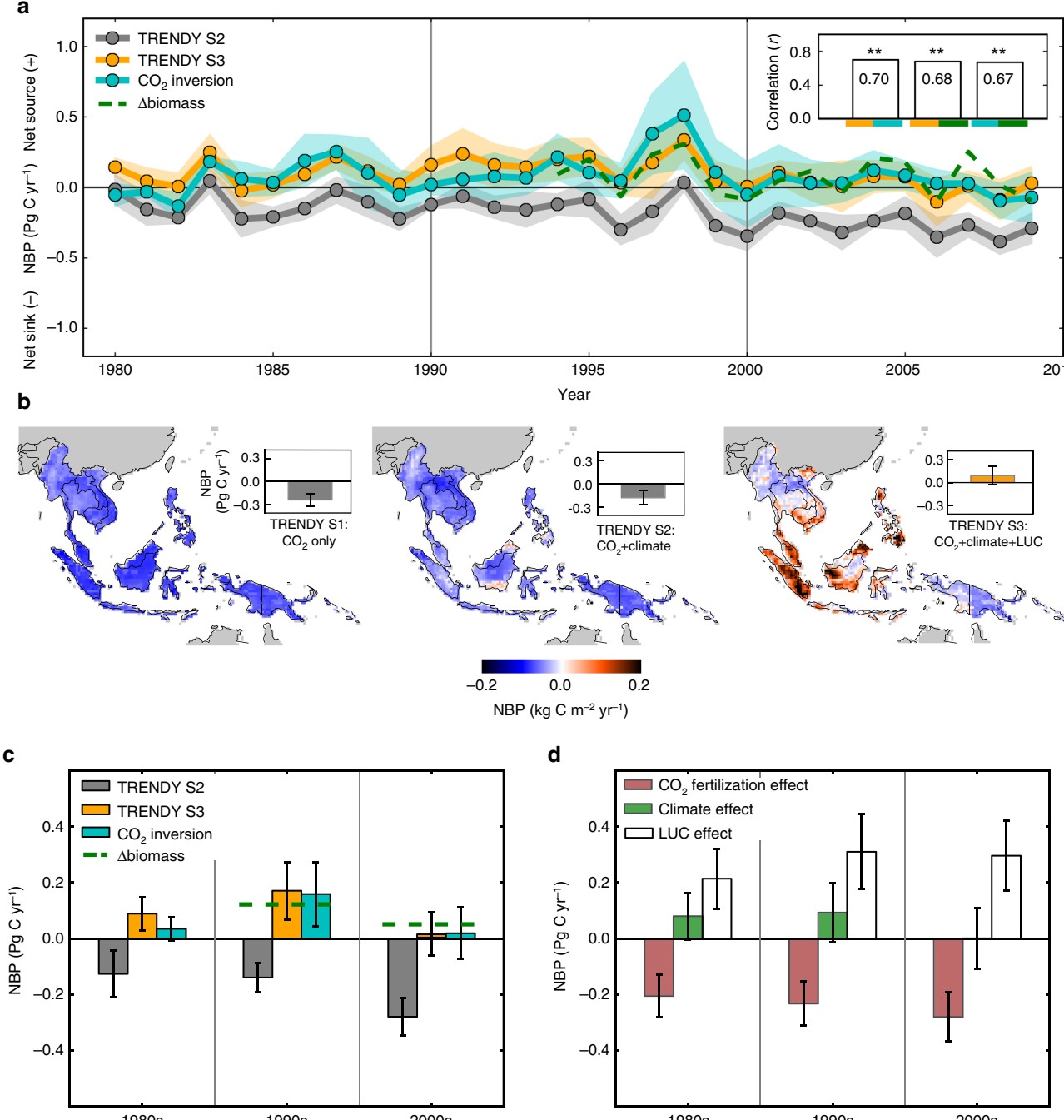

**Fig. 1** Interannual and decadal variability of net $CO_2$ flux in Southeast Asia for 1980–2009. **a** Interannual variability of ensemble averaged NBP from the TRENDY (grey: TRENDY S2; orange: TRENDY S3) and atmospheric $CO_2$ inversions (cyan) for the period 1980–2009, and annual biomass change (dashed green line: Δbiomass) for the period 1994–2009. Shading for the TRENDY and atmospheric $CO_2$ inversions represents 1σ variation among models. A top-right panel shows correlation coefficients (*r*) between interannual variability of the three NBP estimates for the overlapping periods (1980–2009 for the TRENDY and atmospheric $CO_2$ inversions; 1994–2009 for the TRENDY and Δbiomass, and for the atmospheric $CO_2$ inversions and Δbiomass) and statistical significance is indicated by **$p < 0.01$. Negative values in NBP represent a net sink, and positive values a net source. **b** Spatial variability of mean annual NBP from the TRENDY (seven model ensemble average) for the period 1980–2009. Results are shown for the three simulations: forced with varying $CO_2$ only (left: TRENDY S1); varying $CO_2$ and climate (middle: TRENDY S2); and varying $CO_2$, climate and LUC (right: TRENDY S3). Bar graphs represent mean annual NBP by the TRENDY simulations (grey: TRENDY S1 and S2; orange: TRENDY S3) for the period 1980–2009 with error bars representing 1σ variation among models. **c** Decadal NBP budgets from the TRENDY (grey: TRENDY S2; orange: TRENDY S3) and atmospheric $CO_2$ inversions (cyan) for the 1980s, 1990s and 2000s, with error bars representing 1σ variation among models. Decadal budgets from annual biomass changes are shown with dashed horizontal lines for the 1990s (1994–1999) and 2000s (2000–2009). **d** Decadal variability of the attributing factors to NBP from the TRENDY (crimson: the $CO_2$ fertilization effect, green: the climate effect and white: the LUC effect) with error bars representing 1σ variation among models

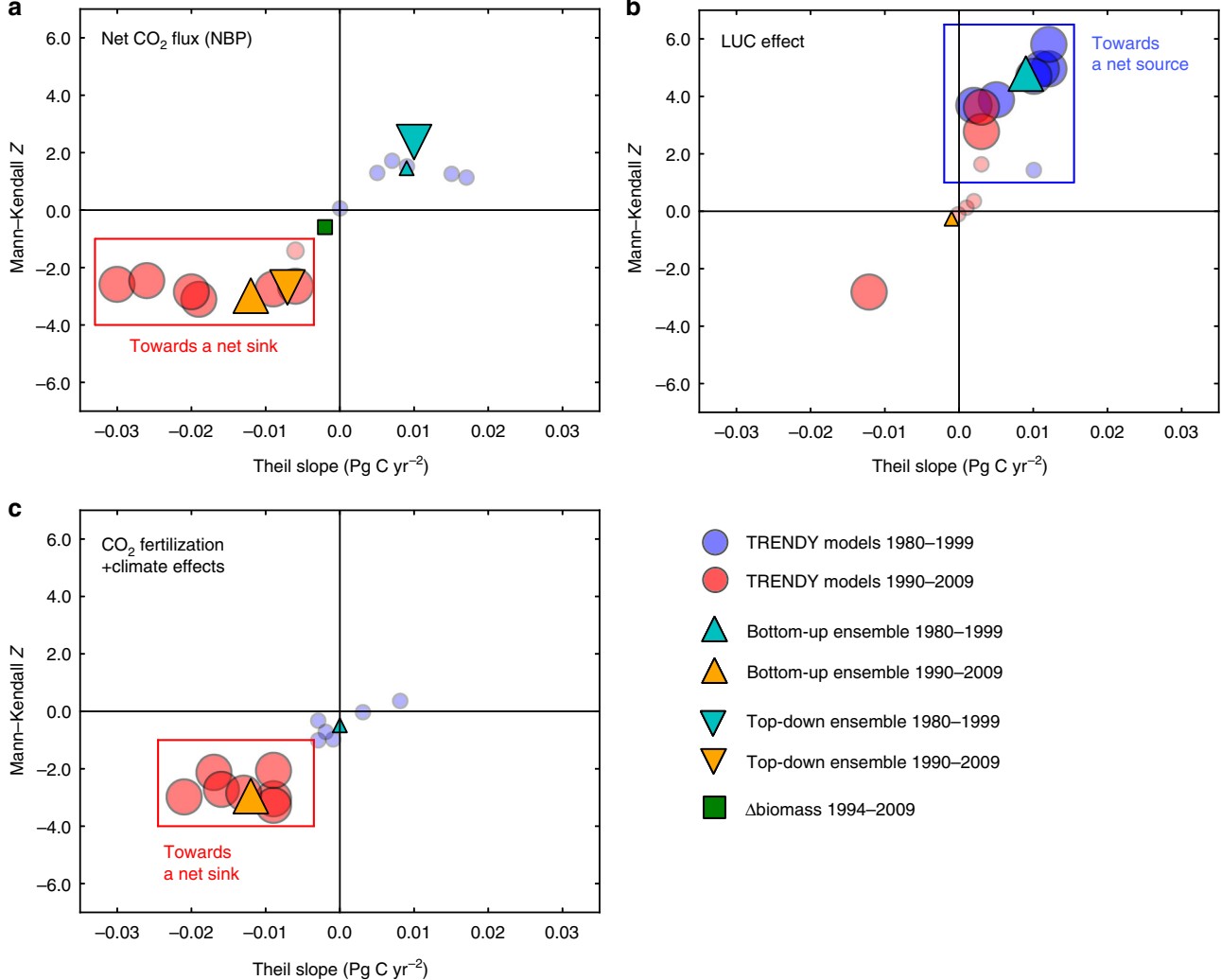

**Fig. 2** Trends in net $CO_2$ flux and its components for the past 30 years. Results of two trend tests (Mann–Kendall and Theil slope tests) on **a**, net $CO_2$ flux (NBP) from the TRENDY S3 (seven models: blue circles for 1980–1999 and red circles for 1990–2009, and ensemble average: a cyan upper triangle for 1980–1999 and an orange upper triangle for 1990–2009) are shown along with those on the atmospheric $CO_2$ inversions (ensemble average: a cyan lower triangle for 1980−1999 and an orange lower triangle for 1990–2009) and Δbiomass (a green square for 1994−2009). The trend tests on attributing factors to NBP are illustrated for **b**, the LUC effect and **c**, $CO_2$ fertilization+climate effects. Size of markers indicates statistical significance of trends: larger ($p < 0.05$) and smaller ($p \geq 0.05$)

and Δbiomass) and components of NBP from the TRENDY model simulations: LUC emissions, fire emissions and plant $CO_2$ exchange (difference between $CO_2$ uptake by photosynthesis and release by plant respiration and decomposition), in relation to variability in the Multivariate ENSO Index (MEI: NOAA ESRL, http://www.esrl.noaa.gov/psd/data/correlation/mei.data). $CO_2$ fluxes between the years that seasonal MEI indicates a moderate-to-strong tendency towards El Niño (hereafter, intense El Niño years; Methods) and the rest of years are compared for the three decades. In the 1980s and 1990s, which are characterized by the occurrence of strong and persistent El Niño events (e.g. 1982/1983, 1987/1988, and 1997/1998; Fig. 3), all the three estimates of NBP show a clear tendency towards a net loss of $CO_2$ from the land in the intense El Niño years compared with the rest of years, with differences amounting to 0.13–0.14 and 0.14–0.26 Pg C yr$^{-1}$, respectively (Fig. 4a, b). In the 2000s, however, no intense El Niño is indicated by MEI (Fig. 3), resulting in a near-neutral carbon balance (Fig. 4c). This result suggests that, in addition to enhanced growth from the $CO_2$ fertilization, the absence of

intense El Niño events is one of the primary causes for the reduced net emission of Southeast Asia in the 2000s.

The investigation of the intense El Niño years revealed that the strength of ENSO largely affects plant $CO_2$ exchange and fire emissions, taking into account peat and deforestation fires (results from Community Land Model (CLM); Methods). In the 1980s and 1990s, $CO_2$ uptake by plants notably shifted towards a net emission by 0.07–0.09 Pg C yr$^{-1}$ in the intense El Niño years when compared with the remaining years (Fig. 4a, b). Likewise, fire emissions by CLM show larger emissions by 0.11 Pg C yr$^{-1}$ in the intense El Niño years than the rest of years in the 1980s and by 0.14 Pg C yr$^{-1}$ in the 1990s. A caveat is that the strength of ENSO had negligible influence on the magnitude of fire emissions simulated without considering the contribution from peat and deforestation fires (an ensemble average excluding CLM). In contrast with plant $CO_2$ exchange and fire emissions, the magnitude of LUC emissions was nearly unchanged regardless of El Niño conditions. This may be expected, because carbon removal by deforestation, and wood and crop harvesting is

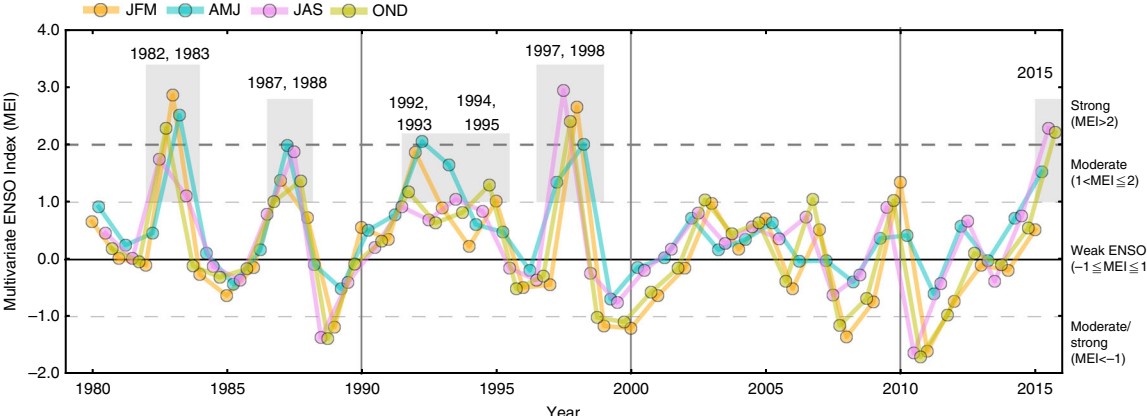

**Fig. 3** Interannual variability in seasonal Multivariable El Niño-Southern Oscillation Index. Interannual variability in seasonal (3-month averaged, i.e. JFM, AMJ, JAS and OND) Multivariable ENSO Index (MEI). Boundaries for the El Niño and La Niña categorization are indicated by dashed lines, and years correspond to the condition for the intense El Niño years are highlighted by grey shadings (see Methods)

triggered by human activities, which do not directly respond to climate conditions (with the possible exception of fire following deforestation).

**Climate sensitivity of $CO_2$ fluxes in Southeast Asia.** Temperature and water availability to plants (indicated by Standardized Precipitation Index: SPI) explained most variability in plant $CO_2$ exchange in Southeast Asia. Among the main meteorological inputs for the TRENDY models (i.e. temperature, precipitation and short-wave radiation) and three types of SPI (based on 3-, 6-, and 9-month moving windows of rainfall accumulation, see Methods), temperature and SPIs are the variables that show a significant association with seasonal variability in plant $CO_2$ exchange: a positive relationship in the former and a negative relationship in the later (Fig. 5a; interannual variability of individual variables in Supplementary Fig. 9). The strong relationships found with SPIs suggest that cumulative precipitation over preceding months is more effective to plant $CO_2$ exchange than simple monthly precipitation, and the empirically upscaled eddy flux data (the FLUXCOM global carbon flux dataset; Methods) confirm these relationships (Fig. 5a). Furthermore, both TRENDY and FLUXCOM data indicate that $CO_2$ uptake by plants decreased due to reduced photosynthesis (Gross Primary Production: GPP) during periods with large increases in temperature and decreases in water availability associated with El Niño (Fig. 5b, c, and Supplementary Fig. 10). The smaller spread in temperature and SPI anomalies during 2000–2009, compared to the 1980–1999 period, implies that weak ENSO variability during the 2000s sustained the high carbon assimilation capacity of plants in Southeast Asia.

In addition to $CO_2$ fluxes from plant ecosystems, our results emphasize fire emissions as an apparent contributor to net $CO_2$ flux resulting from severe droughts[17]. Particularly, contrasting fire emissions between the intense El Niño years and other years highlights the importance of peat and deforestation fire emissions in the carbon balance of Southeast Asia[18] (Fig. 4). The occurrence of strong fire emissions indicated by CLM is found in the years of negative precipitation anomalies corresponding to the intense El Niño years (Supplementary Fig. 11), which is consistent with fire emissions estimated by the Global Fire Emissions Database version 4.1s (GFED4.1s)[19] and reports from remote-sensing-based and model-based studies[7, 20, 21]. However, even with the emissions from peat and deforestation fires, $CO_2$ emissions in the intense El Niño years are still considered as an underestimation of the reality because the TRENDY models, including CLM, do not consider emissions from peat oxidative decompositions following

peat fire events, which could promote even larger $CO_2$ emissions during El Niño events[22].

## Discussion

Our results indicate that a synthesis of multiple approaches (i.e. top-down, bottom-up and remote-sensing-based approaches) is an effective method to constrain regional carbon balance and to elucidate causes for major changes in a projection of $CO_2$ fluxes. The implementation of LUC processes to biosphere models is required to estimate net $CO_2$ flux in Southeast Asia, as indicated by agreement in the interannual and decadal variability of net $CO_2$ flux between the biosphere models, atmospheric $CO_2$ inversions and Δbiomass, which has not addressed in the tropical regions before this study. Our analysis provides new insights for reconciling the top-down and bottom-up regional fluxes over the tropical regions, where wide gaps were reported in the previous IPCC assessment[15], and also serves as a useful precedent for future regional carbon balance assessments by REgional Carbon Cycle Assessment and Processes (RECCAP)[23].

The strength of ENSO exerts a strong control on the carbon balance of Southeast Asia, causing the unique variability in the decade of 2000s. Along with the recent enhancement of $CO_2$ fertilization effect[24], we showed that a milder, less variable, climate due to the absence of intense El Niño events contributed to the reduction of $CO_2$ emissions between the 1990s and 2000s. A recent synthesis of regional emissions of the greenhouse gases (GHG) indicates that Southeast Asia is characterized by the largest emissions not only of $CO_2$ from ecosystems but also of agricultural methane and nitrous oxide among the world regions[25]. Our results suggest that the land response to weak natural climate variability could serve as a strong mitigation to $CO_2$ emissions, even for the world largest ecosystem GHG emitter.

One aspect not addressed here is the role of La Niña (the opposite phase to El Niño) in the decadal NBP shift. La Niña can also induce a milder climate condition, which in turn occasionally enhances regional $CO_2$ uptake[26–28]. We emphasize the significance of El Niño events in the Southeast Asian carbon balance because of a difference in the impact on tropical $CO_2$ fluxes between the two climate phenomena[29]. As illustrated in Fig. 6a, b, linear relationships between seasonal MEI and NBP anomaly (based on the ensembles of the TRENDY models and atmospheric $CO_2$ inversions; individual model results in Supplementary Figs. 12, 13 and Supplementary Table 4) demonstrate that moderate and strong El Niño events (i.e. MEI >1) in the 1980s and 1990s are directly related to a large net source by the land

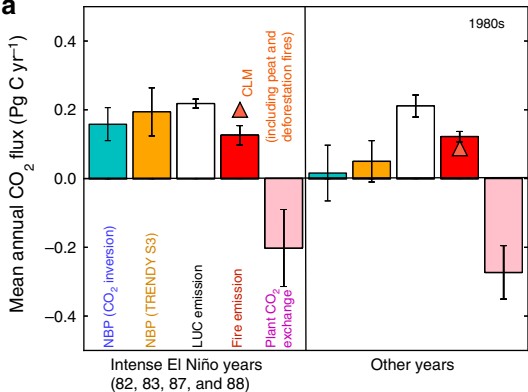

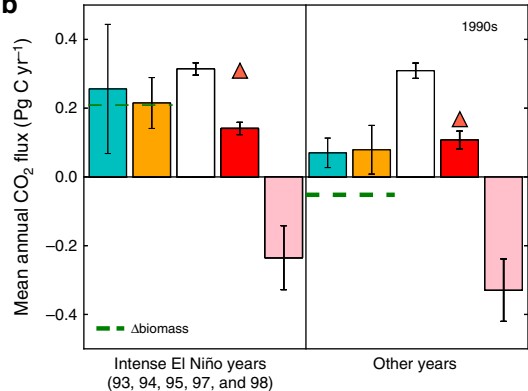

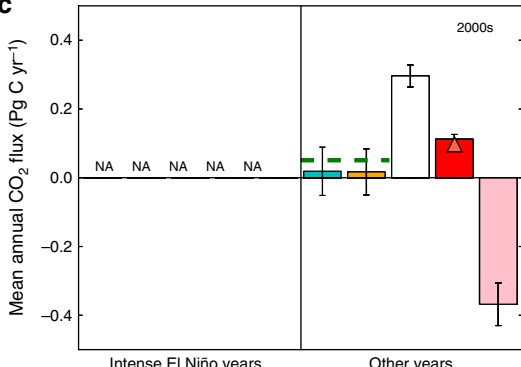

**Fig. 4** Influence of the intense El Niño years on annual $CO_2$ fluxes of the past three decades. Comparison of mean annual $CO_2$ fluxes (NBP, LUC emissions, fire emissions and plant $CO_2$ exchange) between the intense El Niño years and rest of years for **a** the 1980s, **b** the 1990s and **c** the 2000s. Mean annual NBP from the atmospheric $CO_2$ inversions (cyan) and TRENDY S3 (orange) are shown for the three decades, and Δbiomass (dashed green horizontal line) for the 1990s (1994–1999) and the 2000s (2000–2009). Component fluxes such as LUC emissions (white) and fire emissions (red), and plant $CO_2$ exchange (pink) are the estimates from the TRENDY S3. Fire emissions considering the attribution from peat and deforestation fires (the CLM model; orange triangles) are shown separately from an ensemble average of fire emissions by the other models. Error bars of each flux represent 1σ variation among models

biosphere. In contrast, the data corresponding to La Niña and weak ENSO events tend to cluster around carbon neutral, implying that $CO_2$ uptake during La Niña events (such as years 1984/1995, 1989/1990, and 1989/1990) is less significant compared to $CO_2$ emissions during El Niño events. Importantly, in the MEI–NBP relationship, a tendency towards a net source disappeared during the 2000s (Fig. 6c). The 2007/2008 La Niña is

one of the strongest events during the past 30 years (Fig. 3). However, its contribution to the decadal carbon balance shift in the 2000s is incomparable to that from the absence of moderate and strong El Niño events (Fig. 6e). Given the response of the land biosphere to El Niño events, we suggest that the dominant driver of the NBP shift in the 2000s is the absence of moderate and strong El Niño events, with La Niña events playing a lesser role.

The absence of moderate-to-strong El Niño events in the 2000s was a unique case where a natural climate cycle acted to mitigate $CO_2$ emissions in Southeast Asia, albeit only for a limited period. Our estimates of NBP for the period 2010–2016 (see Methods) indicate net $CO_2$ emissions comparable to those in the 1980s and 1990s during El Niño events after 2009 such as the 2015/2016 El Niño (Fig. 6d). An assessment of climate model simulations suggest that surface ocean warming over the eastern equatorial Pacific may lead to an increase in the frequency of intense El Niño events in the future[30], reducing the likelihood that natural climate cycles offer mitigation of $CO_2$ emissions, and therefore leading to stronger positive carbon-climate feedback. In the long term, national-level efforts for forest conservation and ecosystem management, such as the Reduce Emissions from Deforestation and forest Degradation (REDD+) project, are critical to protecting the current $CO_2$ sink capacity of the Southeast Asia[31, 32].

## Methods
**Sign convention for net $CO_2$ flux.** In this study, we chose the sign convention for net $CO_2$ flux that is commonly used in top-down analyses: the negative sign (−) for a net sink and the positive sign (+) for a net source. This sign convention is consistently used throughout the analysis regardless of the TRENDY models, atmospheric $CO_2$ inversions and annual biomass change, and it is also applied not only to NBP but also to plant $CO_2$ exchange. It should be noted that a common sign convention for these variables in bottom-up analyses are opposite to this study[33].

**Bottom-up net $CO_2$ flux.** Outputs from the TRENDY model intercomparison project version 2 (TRENDY)[34, 35] were used to calculate the bottom-up net $CO_2$ flux. Simulations of the biosphere models that participated in the TRENDY were prepared with a consistent forcing dataset: (1) atmospheric $CO_2$ concentration for 1860–2012 based on ice-core measurements and stationary observations from NOAA, (2) climate dataset for 1901–2012 based on a merging between Climate Research Unit (CRU) TS3.2 0.5° × 0.5° monthly climate data[36] and National Centers for Environmental Prediction (NCEP) and National Center for Atmospheric Research Reanalysis 2.5° × 2.5° 6-hourly climate data[37], and (3) 0.5° × 0.5° gridded annual LUC dataset for 1860–2012[38].

The TRENDY models were simulated under three protocols: a protocol that considers variability in atmospheric $CO_2$ (TRENDY S1); a protocol that considers variability in $CO_2$ and climate (TRENDY S2); and a protocol that considers variability in $CO_2$, climate and historical LUC (TRENDY S3). For each protocol, the models first established an equilibrium state of carbon balance by a spin-up run, which is forced with the 1860 $CO_2$ concentration (287.14 ppm), recycling climate mean and variability from the early decades of the twentieth century (i.e. 1901–1920) and constant 1860 crops and pasture distribution. Then, simulations for two transient periods were conducted. For the period 1861–1900, the models were forced with varying $CO_2$ concentration and recycling climate (as in spin-up) in the TRENDY S1 and S2, and in addition varying LUC in the TRENDY S3. After the 1861–1900 period, the models were consecutively run for the 1901–2012 period with varying $CO_2$ concentration and recycling spin-up climate in the S1, varying $CO_2$ concentration and climate in the TRENDY S2 and S3, and varying LUC in the TRENDY S3. A summary of the forcing data configuration for these simulations is shown in Supplementary Fig. 2.

Among the participating models, we selected seven models that satisfy necessary criteria for the analyses such that models provide monthly NBP outputs for all three simulations and an explicit output of annual LUC emissions. Specifically, they are the CLM version 4.5[39], Integrated Science Assessment Model[40], Joint UK Land Environment Simulator (JULES) version 3.2[41], Lund-Potsdam-Jena DGVM wsl (LPJ)[42], LPJ-GUESS[43], Orchidee-CN (O-CN)[44] and Vegetation Integrative SImulator for Trace gases (VISIT)[45]. Spatial resolutions of the TRENDY model outputs are not consistent among these seven models; fine resolutions were used in some models and coarse resolution in others (Supplementary Table 1). For models whose outputs were submitted with coarse spatial resolution, we rescaled grids so that all seven model outputs have the consistent spatial resolution of 0.5° × 0.5°.

**LUC emissions**. The LUC forcing for the TRENDY models provides gridded historical transitions of land use, based on annual changes of cropland and pastureland area, and wood harvest from the UN Food and Agricultural Organization (FAO) national statistics. Historical changes in annual area of cropland and pastureland were determined by the HistorY Database of the global Environment (HYDE) model version 3.1[46], which takes the FAO national statistics for cropland and pastureland as the main input source, and spatializes the statistics at the spatial resolution of $5' \times 5'$ using allocation algorithms and time-dependent weighting

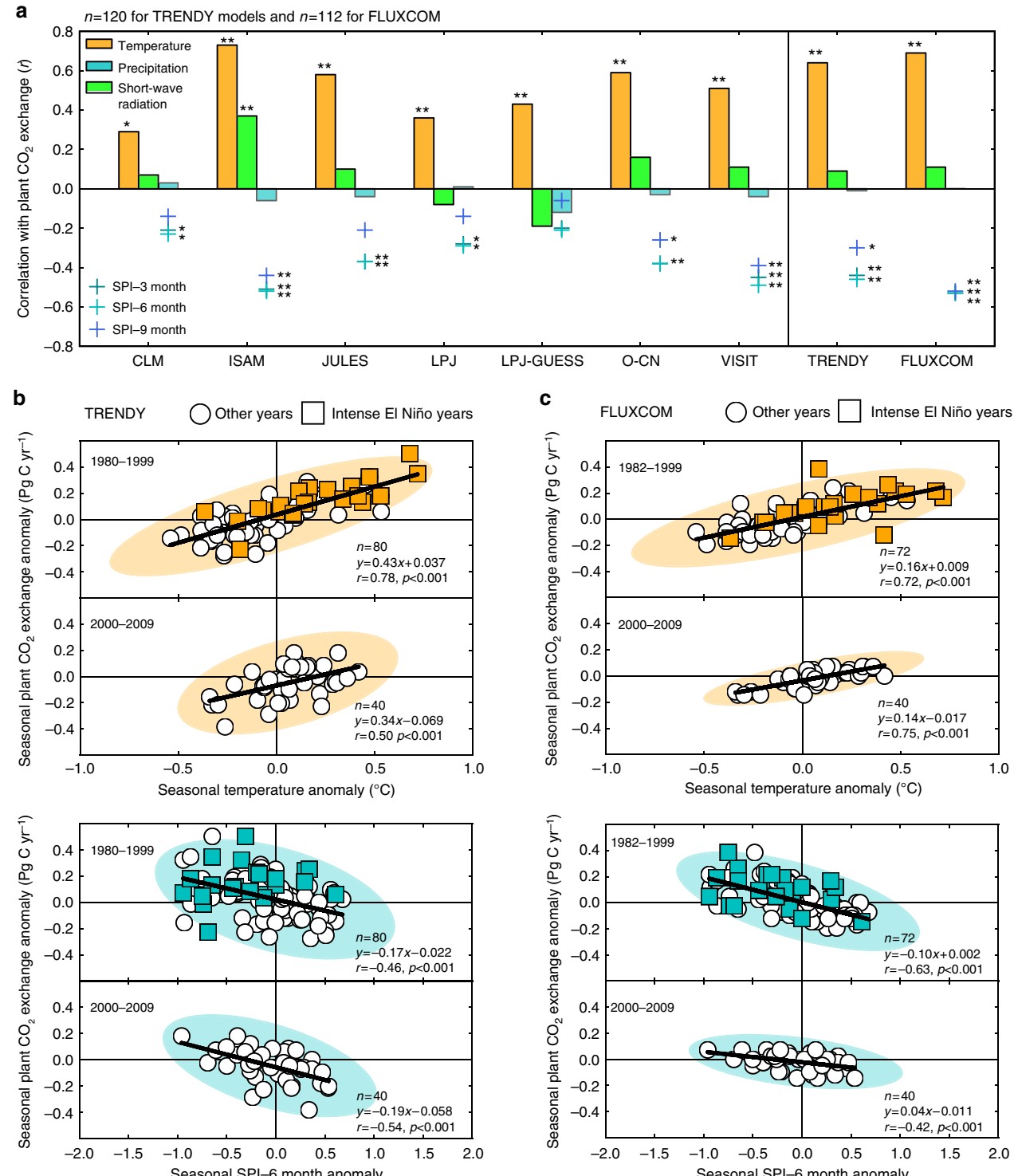

**Fig. 5** Climate sensitivity of seasonal plant $CO_2$ exchange by the biosphere models and empirical upscaling. **a** Correlation coefficients ($r$) in relationship between seasonal anomalies of plant $CO_2$ exchange induced by the climate effect (TRENDY S2–S1) and climate variables (temperature, precipitation, short-wave radiation and three types of SPIs) for the period 1980−2009. Results are shown for seven individual TRENDY models and the ensemble of the TRENDY and FLUXCOM data. Statistical significances are indicated by **$p < 0.01$ and *$p < 0.05$. Relationships between seasonal anomalies of plant $CO_2$ exchange and temperature, and SPI–6 months for the periods 1980–1999 and 2000–2009, for **b** the TRENDY and **c** FLUXCOM. All relationships are shown along with the 95% confident ellipses and linear regressions. Square markers indicate data corresponding to the intense El Niño years and circle markers to the rest of years. All relationships with SPI are calculated as SPI leads plant $CO_2$ exchange by 3 months

maps based on global historical population density, soil suitability, distance to rivers, lakes, slopes and biome distributions. The HYDE cropland and pastureland status were then combined with the wood harvest status based on the FAO national wood harvest statistics in order to extend global land use patterns, including transitions of cropland, pastureland, primary and secondary lands (an extended version of HYDE)[38]. First, the gridded cropland and pastureland area from the HYDE model was rescaled from $5' \times 5'$ to $0.5° \times 0.5°$ resolution, and at the same

time, fractions occupied by cropland and pastureland was calculated for each rescaled grid cell. By subtracting fractions of cropland and pastureland (and water/ice if any) from each grid cell, fractions of natural vegetation (primary or secondary lands) was also determined for each grid cell. Distinction between primary and secondary lands (previously disturbed by human activities or not) and fractions of these land types occupied in each grid cell were determined based on the spatialized FAO wood harvest data with empirically estimated biomass density maps produced

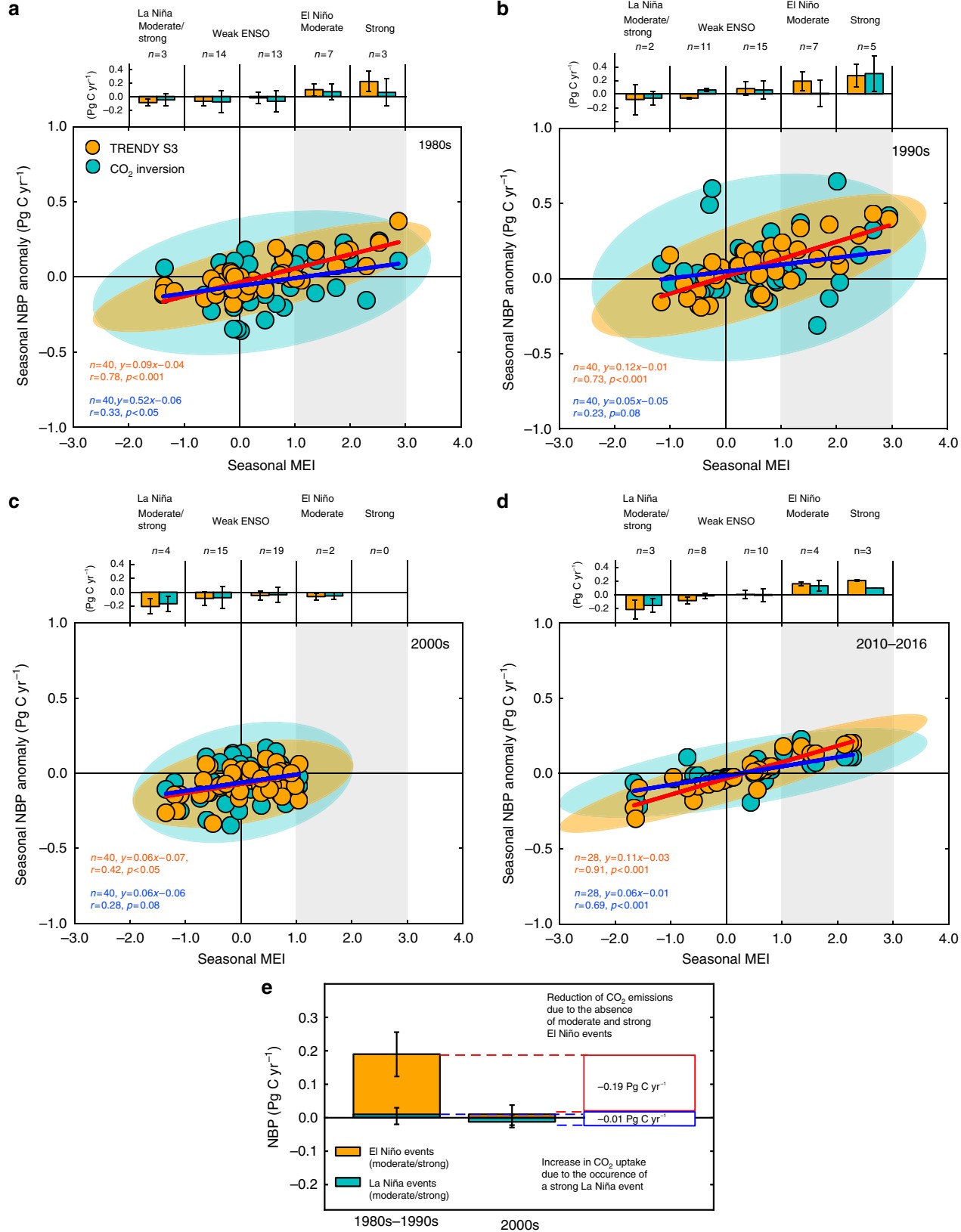

at the spatial resolution of 0.5° × 0.5° from Miami-LU model[47]. Both the HYDE and extended HYDE models assume a strong association between land use and human population[48]. Thus, interannual variability of the land use status for the past-1960 period (prior to availability of the FAO statistics) is mainly induced by historical population density. Decadal changes in fractions occupied by cropland, pastureland, primary and secondary lands for Southeast Asia by the extended HYDE data are shown in Supplementary Fig. 5.

LUC emissions in the TRENDY models account for the net effect of LUC on terrestrial carbon cycle including instantaneous and legacy emissions. In each model, forest area changes (deforestation or afforestation) in response to annual changes in cropland and pastureland area predefined by the forcing data, resulting in a relatively consistent forest area changes due to LUC among the models (minor differences occur due to dynamic vegetation). However, specific schemes for LUC modelling are left to the discretion of each modelling group, which means that fundamental assumptions and levels of complexity in LUC modelling vary among the models: for instance, distinction of primary and secondary forests, implementation of wood and crop harvests, consideration of residue carbon after deforestation and turnover rates of a product pool (Supplementary Table 1). These different schemes of LUC modelling induce non-negligible variations in estimates of LUC emissions upon close examination. Thus, application of LUC emissions by the TRENDY is limited to long-term and regional-scale analyses, which aim to capture strong signals or trend shifts of $CO_2$ uptake or release. Further details of the LUC modelling in the TRENDY are provided in ref. [35], and a comprehensive comparison of LUC emissions by the TRENDY and other independent assessments in Asia is conducted in ref. [49].

**Fire emissions.** Among the selected seven models, CLM, LPJ, LPJ-GUESS and VISIT provided outputs of fire emissions. Despite differences in details, modelling for global fire emissions by those four models are based on similar schemes, which primarily depend on amounts of fuel load (e.g. vegetation, litte and woody debris) and moisture availability in litter, soil or near-surface air[50–52]. For Southeast Asia, CLM provides more realistic variability in fire emissions than others because the model considers the contribution of peat and deforestation fires (Supplementary Fig. 11)[53].

In CLM, simulations of peat and deforestation fire emissions are processed first by the calculation of their burnt area. The burnt area due to peat fires is estimated by considering effects of climate and inundation of peatlands with a gridded static map (0.5° × 0.5°) of peatland from ref. [54], and that due to deforestation fires by considering effects of climate and deforestation rates represented by decreased tree coverage fractions from the land use data[38, 55]. Subsequently, fire emissions are calculated by applying the estimated burnt area, fuel load and functional type (PFT)-dependent combustion completeness factors. These simulation results have been validated against the observed interannnual variability of peat fires from ref. [7] and the GFED 3 burnt area and fire emission products[56].

**Attributions to net $CO_2$ flux.** Effects of $CO_2$, climate and LUC on NBP were isolated by the manipulation of the TRENDY S1, S2 and S3 by following work of refs [57, 58]. The $CO_2$ fertilization effect is represented by NBP of the TRENDY S1, because only $CO_2$ concentration varies in the TRENDY S1. The climate effect was extracted by subtracting NBP of the TRENDY S1 from that of the TRENDY S2 (S2–S1). Because the TRENDY S1 considers variability in $CO_2$ and the TRENDY S2 considers variability in $CO_2$ and climate, their difference leaves out the effect of $CO_2$ fertilization and only the effect of climate remains. Similarly, the LUC effect was extracted by subtracting NBP of the TRENDY S2 from that of the TRENDY S3 (S3–S2); their difference leaves out the effects of $CO_2$ fertilization and climate, and only the effect of LUC remains. The climate effects on plant $CO_2$ exchange, GPP and ecosystem respiration (RE) were calculated by the above-mentioned approach (subtracting results of the TRENDY S1 from those of the TRENDY S2), except that plant $CO_2$ exchange, GPP and RE were used in place of NBP.

**Top-down net $CO_2$ flux.** Top-down net $CO_2$ flux is represented by five atmospheric $CO_2$ inversions: ACTM v5.7b (ACTM)[59], JENA s81 v3.8 (JENA)[60], JMA-CDTM (JMA)[61], MACC v14r2 (MACC)[62], and NICAM-TM (NICAM)[63]. These models estimate net $CO_2$ flux by the inversion of continuous and discrete atmospheric $CO_2$ measurements from global networks (e.g. NOAA Earth System Research Laboratory (NOAA/ESRL), World Data Centre for Greenhouse Gases

(WDCGG), Comprehensive Observation Network for TRace gases by AIrLiner (CONTRAIL) and GLOBALVIEW) with prior fluxes (land and ocean fluxes, fire emissions and anthropogenic $CO_2$ emissions). These inversions minimize a Bayesian objective function with an assumption that errors form a Gaussian distribution, and error correlation is represented by off-diagonal elements in the posterior error covariance matrix. A choice of $CO_2$ measurements and prior fluxes for each inversion system was left to the discretion of modelling groups, as well as spatial resolution and time period of inverted fluxes (Supplementary Tables 2, 3). Top-down net $CO_2$ flux for 1980–2009 was estimated by an ensemble average of inversions for overlapping time periods (i.e. JENA and MACC for 1980–1984; JENA, JMA and MACC for 1985–1987; JENA, JMA, MACC and NICAM-TM for 1988–1989; ACTM, JENA, JMA, MACC and NICAM-TM for 1990–2007; ACTM, JENA, JMA and MACC for 2008–2009).

**Satellite-based annual biomass change.** We used the satellite-based gridded (0.25° × 0.25°) global aboveground biomass covering the period 1993–2012 (Global Aboveground Biomass Carbon version 1.0) to estimate the annual biomass changes (Δbiomass)[64]. The global aboveground biomass is estimated based on harmonized vegetation optical depth (VOD) data derived from multiple passive microwave satellite sensors, including Special Sensor Microwave Imager, Advanced Microwave Scanning Radiometer for Earth Observation System, FengYun-3B Microwave Radiometer Imager and Windsat. The obtained VOD data are converted to aboveground biomasses via an empirical relationship between the VOD data and satellite-based spatial map of aboveground biomass for tropical regions[65]. The global distribution of total biomass is estimated by applying conversion factors, obtained from literatures for different forests and non-forest vegetation, to the aboveground biomass data. We calculated Δbiomass by simply taking differences between the total biomass data of current and preceding years for each grid cell for the period 1994–2009, and aggregated the grid data for the Southeast Asia region.

**Empirical upscaling of eddy flux data.** We used the empirical upscaling of eddy flux observations to compare against climate sensitivity of $CO_2$ fluxes by the TRENDY model simulations. The FLUXCOM global carbon flux dataset[66, 67] is an ensemble of daily carbon fluxes estimated from machine learning algorithms (Random Forest[68], Artificial Neural Network[69] and Multivariate Adoptive Regression Splines[70]) trained with 224 eddy flux tower observations and climate data. The training of the three machine learning algorithms were conducted separately for GPP and RE with explanatory variables with spatial (e.g. plant functional type), spatial and seasonal (e.g. mean seasonal variations of land surface temperature, vegetation index) and spatial, seasonal and interannual (e.g. climate variables) variations were used. Using the trained machine learning algorithms and spatial input data, MODIS product (with no interannual variations) and climate variables (with interannual variations) from CRUNCEPv6 (http://esgf.extra.cea.fr/thredds/catalog/store/p529viov/cruncep/V6_1901_2014/catalog.html), spatio-temporal GPP and RE were forced with grids of 0.5° × 0.5° spatial resolution and daily time step for the period 1980–2013. Subsequently, spatiotemporal variability of plant $CO_2$ exchange was calculated by mass balance from the upscaled GPP and RE products (i.e. GPP-RE).

In the analysis, we compared $CO_2$ fluxes induced by the climate effect (TRENDY S2-S1) against the FLUXCOM because $CO_2$ fluxes from the FLUXCOM are results of upscaling natural vegetation fluxes, which make ideal products to evaluate regional climate sensitivity of $CO_2$ fluxes from plants. It should be noted that the FLUXCOM data do not account for $CO_2$ losses from LUC because no predictor variables about LUC were used in its estimation.

**Tests for significance of decadal trend of net $CO_2$ flux.** We applied the two commonly used non-parametric tests for the slope in linear regression, Mann–Kendall and Theil slope tests[71, 72], for the detection of robust trends in NBP and its attributing factors (i.e. the $CO_2$ fertilization, climate and LUC effects). Mann–Kendall test takes a list of data ordered in time and calculates test statistics (i.e. Mann–Kendall Z), in which takes the number of positive and negative differences between paired data and normalizes it by a square root of its variance. Theil slope calculates the slope of linear regression as the median of all slopes between paired values of data of interest.

The trend tests are conducted for the periods 1980–1999 and 1990–2009 to multiple estimates of annual NBP: (1) an ensemble average of the atmospheric $CO_2$

**Fig. 6** Decadal patterns of relationships between El Niño-Southern Oscillation and net $CO_2$ flux anomaly. Relationship between seasonal MEI and NBP anomaly from the TRENDY S3 (orange) and atmospheric $CO_2$ inversions (cyan) for **a** the 1980s, **b** 1990s, **c** 2000s and **d** current period (2010–2016). MEI and NBP anomaly are 3-month averaged (i.e. JFM, AMJ, JAS and OND), and their relationships are constructed in such a way that MEI leads the NBP anomaly by 3 months to account for the observed lag of influence by El Niño on $CO_2$ fluxes (see Methods). Along with scatter plots, 95% confident ellipses and regression lines are shown for the TRENDY S3 and atmospheric $CO_2$ inversions. Grey shading represents ranges of large positive MEI values and positive NBP anomalies. Bar graphs on the top of the scatter plots are seasonal NBP anomaly averaged for different strengths of ENSO; MEI < −1 (moderate and strong La Niña), MEI = −1 to 1 (weak ENSO events), MEI = 1 to 2 (moderate El Niño), and MEI > 2 (strong El Niño). Error bars represent 1σ variation of data within different strengths of ENSO. **e** Budgets of NBP by the TRENDY S3 corresponding to moderate/strong La Niña and El Niño events in the decades of 1980s–1990s and 2000s. Error bars represent 1σ variation among models

inversions, (2) an ensemble average and (3) individual outputs of the seven models from the TRENDY, and (4) Δbiomass for the period 1994–2009 (Fig. 2a). For trends of the attributing factors to NBP, we take the combined effect of $CO_2$ fertilization and climate (NBP from the TRENDY S2) and the LUC effect (difference in NBP between the TRENDY S3 and S2) from the TRENDY model simulations, and applied the tests to an ensemble average and seven individual model outputs (Fig. 2b, c).

**Condition for intense El Niño years**. We categorized the conditions for El Niño and La Niña years based on seasonal variability in the MEI obtained from US National Oceanic and Atmospheric Administration (NOAA: http://www. esrl.noaa.gov/psd/data/correlation/mei.data). First, a simple categorization is conducted under a rule that seasonal MEI (i.e. 3-month averages, JFM, AMJ, JAS and OND) falls within the predefined range at least one season of year, such that MEI <−1 (strong/moderate La Niña), −1 ≤ MEI ≤ 1 (weak ENSO), 1 < MEI ≤ 2 (moderate El Niño) and MEI > 2 (strong El Niño). To characterize the intensity of El Niño events, however, not only the magnitude but also the duration of seasonal MEI needs to be considered. Therefore, we then defined Intense El Niño years, which refer to years that seasonal MEI values falls with MEI >1 (moderate or strong El Niño) at least for two seasons (see Fig. 3). We excluded the year 1992 from the analyses because the forcing data of the TRENDY do not account for the effect of volcanic aerosol by the Mount Pinatubo eruption on radiation[73, 74].

**Standardized precipitation index**. SPI is an indicator for conditions of dryness and wetness at a given time scale and location of interest based on historic precipitation data[75]. Calculation of SPI is based on cumulative precipitation data for a moving window of different length of months such as 1, 3, 6, 9 months, and so on. Then, the data are fitted to a gamma distribution with parameters $\alpha$ and $\beta$, turning a cumulative precipitation distribution into a probability distribution. Resulting SPI values indicate severity of wetness and dryness, with positive values indicating higher probability of wet events and negative values indicating the opposite. Interpretation of SPI differs by the length of accumulation periods, such that shorter periods (e.g. 1 to 3 months) indicates changes in land surface water and longer periods (e.g. 6 to 9 months) indicates changes of water reservoir.

In this study, we calculated three types of SPI (3, 6 and 9 months) using the CRU-NCEP precipitation that was used as a forcing of TRENDY models.

**Relationship between seasonal MEI and NBP anomaly**. Relationship between seasonal MEI and seasonal NBP anomaly (i.e. 3-month averages, JFM, AMJ, JAS and OND) for the study period (1980–2009) is constructed by considering a lag effect such that MEI leads NBP anomaly by 3 months (Fig. 6; Supplementary Figs. 12, 13) because a majority of the NBP estimates (i.e. the TRENDY models and atmospheric $CO_2$ inversions) yields an optimal correlation at the 3-month lag. Some models show an optimal correlation at the 6-month or 9-month lag, but we regard that lag longer than 3 months is not the best representation of inter-connection between ENSO and terrestrial carbon cycle (Supplementary Table 4).

For the current period (2010–2016: Fig. 6d), we extended the relationships using the TRENDY S3 and atmospheric $CO_2$ inversion data from 2010 to 2012 (2012 is the end of the simulation period of the TRENDY) and then we supplemented the data for the period 2013–2016 by empirical relations between seasonal MEI and ensemble average NBP anomaly with the 3-month lag (a base period 1980–2009) for both the TRENDY S3 and atmospheric $CO_2$ inversions (see results for ensemble averages in Supplementary Figs. 12, 13). Temporal coverages of two atmospheric $CO_2$ inversions (i.e. JENA and MACC) extends beyond the year of 2012 (Supplementary Table 2); however, we chose the consistent method and temporal coverage for empirical regressions for both the TRENDY models and atmospheric $CO_2$ inversions.

**Data availability**. The TRENDY-v2 data are available via Dr. Stephen Sitch, Exeter University (s.a.sitch@exeter.ac.uk). Global Aboveground Biomass Carbon version 1.0, MACC and JENA inversion data are available from the web sites (Above-ground Biomass Carbon: http://www.wenfo.org/wald/global-biomass/, MACC: http://apps.ecmwf.int/datasets/data/macc-ghg-inversions/, JENA: http://www.bgc-jena.mpg.de/CarboScope/). FLUXCOM data, ACTM, JMA and NICAM inversions are available by contacting Drs. Martin Jung (mjung@bgc-jena.mpg.de), Prabir K. Patra (prabir@jamstec.go.jp), Takashi Maki (tmaki@mri-jma.go.jp) and Yosuke Niwa (yniwa@mri-jma.go.jp), respectively.

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

## Acknowledgements

This research was supported by Environment Research and Technology Development Funds from the Ministry of the Environment of Japan (2-1401) and the Environmental Restoration and Conservation Agency (2-1701) and Asia-Pacific Network for Global Change Research (APN: grant#ARCP2011-11NMY-Patra/Canadell). We acknowledge Dr. Tomo'omi Kumagai for providing information about the current status of land use change in Southeast Asia and Mr. Yuji Yanagi for pre-processing the TRENDY model outputs for the analyses. This work is dedicated in memory of Dr. Rikie Suzuki who was the devoted leader of Department of Environmental Geochemical Cycle Research at the Japan Agency for Marine-Earth Science and Technology.

## Author contributions

M.K. designed the analyses with support from K.I., P.K.P., J.G.C., B.P. and L.C. M.K. conducted the analyses, prepared figures and tables, and organized the manuscript. P.K.P., B.P., S.S., Y.Y.L., A.I.J.M.v.D., T.S., P.F., A.A., A.H., A.K.J., E.K., C.K., F.L., T.A.M.P., S.Z., A.W., F.C., T.M., T.N., Y.N. and C.R. contributed data to the analyses. K.I., P.K.P., J.G.C., B.P., S.S., L.C., Y.Y.L., A.I.J.M.v.D., T.S., F.L., F.C. and N.S. contributed to the writing of the manuscript.
