## [Peer Review File(PDF 1038 kb) · Nature Communications]

Reviewers' comments:

Reviewer #1 (Remarks to the Author):

The ms reports on changing CO₂ fluxes from SE Asia since the 1980's, diagnosed from fluxes calculated by 7 "bottom-up" land carbon models, 5 "top-down" flux inversions from atmospheric CO₂ and transport, and aboveground biomass from remote sensing. The finding that there are greater CO₂ fluxes to the atmosphere from fires during El Nino's is not a surprise. It follows that years without El Nino's would not be a strong source.

The finding that temperature anomalies during El Nino years, rather than precipitation anomalies or insolation anomalies, dominate the CO₂ flux anomalies (and GPP anomalies – cf Figure S11) seems counter-intuitive, and brings into question the bottom-up models. SE Asia is typically in drought during El Nino's. Also Nemani et al. (Science 2003) shows that insolation is the dominant control on NPP in the tropics. The authors report the finding, but do not provide a mechanism for such a finding.

I have several concerns about the ms (see below), and regret that I do not find the ms ready for publication in Nature.

Major concerns:

(1) SE Asia is a particularly challenging location for study, as there are few observations to constrain "bottom-up" estimates or "top-down" inversions. Consensus among the models does not mean that the models are correct. I would have appreciated a critical evaluation of the different constraints on the fluxes derived for this region, so that I could have some confidence that the modeling results something have some relevance to the real world.

(2) I am totally perplexed by the comparison with aboveground biomass carbon (ABC) data with the bottom-up and top-down CO₂ fluxes. ABC is an inventory, cross-checked against lidar-based estimates of biomass. I would have thought that ABC would be a metric for cross-checking how the various bottom-up models convert the given landuse areas into ABC. Yet ABC is interpreted as a flux in this ms. I can imagine a drought that diminishes photosynthesis without altering the biomass or killing trees. So the correlations between, and comparisons of, areally-averaged fluxes and areally-averaged ABC may be conflating forested and deforested areas.

(3) The criterion for selecting land carbon models appears to be the availability of NBP output, rather than performance of the models against some metric. IPCC AR5 WG1 Figure 6.17 shows a large range in the sensitivities of (globally averaged) land CO₂ fluxes to temperature and precipitation among the 10 models that participated in the emissions-forced CMIP5 experiments,. Figure 6.17 (IPCC) does not inspire confidence in land carbon models and their projections of changes. Nor does Figure S4 of this ms: LPJ-GUESS has a very heterogeneous response even to CO₂ fertilization – this demands explanation! I find Figures S9-S14 very confusing. See comment above re Figure S11 (GPP). It would have been more instructive to show for each model sensitivities of GPP and RE, rather than NBP, to the climate variables.

(4) "Top-down" inversions typically estimate tropical land fluxes as the residuals to balance the global budget, as there are few observations in the tropics, and the intense convection dilutes the signal. Indeed, Figure S5b shows for the 2000's, three of the 5 models show a source, and two show a sink. The authors should provide constraints on these inversions.

Minor concerns:

(1) The ms is difficult to follow, as the crucial info about the models (e.g. what is simulation 3?) and for assessing the results are in the supplemental info. The authors should remember "Supplemental" means supplemental. The ms text should be able to stand on its own.

(2) I do not know what TRENDY-v2 is about, and do not know what is in the various bottom-up and top-down models. References for the TRENDY-v2 models would have helped.

Reviewer #2 (Remarks to the Author):

This study aims at showing how ENSO, or the neutral-phased ENSO, influences atmospheric CO₂ fluxes in Southeast Asia. The main conclusion is that the 2000s period, when there were no strong El Niños (termed as "neutral-phased ENSO" by the authors), saw a weakened CO₂ source compared to the 80s and 90s as well as in the most recent years. The method is comprehensive and I believe this study would make a useful contribution to the field. However, the paper would benefit from a careful revision to improve its clarity and conviction. To start with, the readers would be confused by the term "neutral-phased ENSO" to refer to the absence of strong El Niño events, rather than the actual neutral events (i.e., neither El Niño nor La Niña events). Secondly, as far as ENSO is concerned, it is odd that there are no discussions on the role of La Niña in CO₂ uptake. For instance, one would wonder how the 2000s period "net sink" is due to the prevalent La Niña events. This issue is not clear because the authors assigned weak/moderate El Niño events, together with La Niña and neutral years, as "non El Niño years". There are also a few places where clarity could be improved as outlined below.

First of all, the paper lacks references in ENSO and its relationship with terrestrial CO₂ cycle. The authors could start with a recent paper by Kim et al. (2016, *J. Climate*, <http://dx.doi.org/10.1175/JCLI-D-14-00672.1>), and earlier studies such as: Hashimoto, H., and Coauthors, 2004: El Niño–Southern Oscillation-induced variability in terrestrial carbon cycling. *J. Geophys. Res.*, 109, D23110, doi:10.1029/2004JD004959.

Jones, C. D., M. Collins, P. M. Cox, and S. A. Spall, 2001: The carbon cycle response to ENSO: A coupled climate–carbon cycle model study. *J. Climate*, 14, 4113–4129.

Rayner, P. J., R. M. Law, and R. Dargaville, 1999: The relationship between tropical CO₂ fluxes and the El Niño–Southern Oscillation. *Geophys. Res. Lett.*, 26, 493–496, doi:10.1029/1999GL900008.

These references can be used in illustrating the different roles between El Niño and La Niña.

With regards to defining ENSO years, then weak/moderate El Niño events should be identified, as well as La Niña events. The analysis will be neater this way, as it will clearly distinguish the effect of El Niño events in general, and importantly providing a proper understanding of the role of ENSO (i.e., La Niña vs El Niño). In the current manuscript, the term "neutral-phased ENSO" to refer to the absence of strong El Niño is confusing, as this term is expected to refer to neutral years. Using the MEI alone is sufficient, i.e., there is no need to add an additional constraint in SOI. Please add a paper in Methods for general definition of ENSO years.

This more refined classification should clarify the rest of the analysis as it should make the difference between El Niño and non El Niño years more significant. For instance the statement in L43: "In the 2000s,...resulting in a weaker CO₂ loss from the land (Fig. 2c)" - is puzzling because Fig. 2c shows no values at all for the El Niño years, i.e., there is nothing to be compared with the 1980s and 1990s. This more refined classification will also enhance the difference between El Niño years (strong and weak El Niños) and non El Niño years (i.e., La Niña and neutral events) within each decade. A supplementary analysis can be added to show the analysis for neutral events only. This will reveal the role of La Ninas. Once all these are clarified, it would be easier to understand what's going in the 2010-2015 period when the models provide incomplete data.

Statistical significance should be noted in Fig. 2, in comparing El Niño years in 2000s versus El Niño years in 1980s and 1990s; as well as in comparing El Niño years vs non El Niño years in each decade.

It is not clear how the 2010-2015 NBP anomalies are obtained, given the absence of full data for that period. In Methods (L547-555), it is stated that the three-month lag correlation for the period 2010-2012 is used to project for the remaining years (2013-2015). But 3 years of data (2010-2012) is not sufficient to represent ENSO. One would need at the very least 10 years to sample ENSO events. Many researchers regard that even 10 years are not long enough. I would also encourage the authors to use available observational data for this recent period rather than just empirical derivation from the models.

L86-88: In contrast to what's stated here, the variations in simulation-2 appear correlated with the atmospheric CO₂ inversions and annual biomass change (Fig. 1a), despite the offset in the mean which should be due to LUC. The corresponding correlations for simulation-2 should also be shown in Fig. 1a to support the statement.

L99 (and elsewhere): "towards a net sink from the 1990s to the 2000s" shouldn't this be "towards a weaker net source from the 1990s to 2000s" as we are looking at the orange bar (simulation-3) which still shows a weak positive value. In fact this should be clearer in Fig. 1d where the net value could also be added as verification.

The sentence in L88-92 is not a good read. What did you mean by "LUC variability (...) changed patterns of the spatial variability"? "..., flux changed" what flux?

Figure 1, caption: "Interannual"

Figure 2: Shouldn't there be an error bar on each of the orange triangles?

L146-158: it should be stated if the differences described here are statistically significant.

L85: "resulting in" should be "as indicated by"

L541: "forwards" should be "leads"

L543-544: "with" should be "at"

Supp. Table 4 caption: indicate the period of analysis (1980-2009?)

Please indicate statistical significance in Supp. Fig. 6.

"Neutral-phased ENSO in the 2000s" sounds quite odd. There are certainly ENSO events in the 2000s, such as the 2002/03, 2006/07 El Niño and 2005/06, 2007/08, and 2008/09 La Niña events.

Reviewer #3 (Remarks to the Author):

In the paper "Neutral-Phased El Niño-Southern Oscillation drives Southeast Asian CO₂ uptake in the 2000s", authors used multiple modelling approaches to estimate CO₂ fluxes and reached the conclusion that "CO₂ release associated with El Niño-induced anomalies were insignificant during the 2000s". Author also suggested that "Strong El Niño events in 2010–2015 indicate that neutral-phased ENSO lasted only during the 2000s and in the future might lead to increased vulnerability of carbon stocks in Southeast Asia."

Authors do not quantify and separate the impact of LUC on CO₂ flux in the past decades from other factors such as ENSO. Therefore, there are not enough evidences to support authors statements, such as "the moderate climate in the 2000s represents a unique case where a natural climate cycle acted to mitigate CO₂ emissions in Southeast Asia"

Reviewer would recommend further analysis to support the final conclusions and findings of this manuscript.

Response to reviewer 1's comments.

Kondo et al.

“Decadal carbon balance shifts controlled by land-use change and El Niño-Southern Oscillation in Southeast Asia” (previous title “Neutral-Phased El Niño-Southern Oscillation drives Southeast Asian CO₂ uptake in the 2000s”)

Major changes are marked as red in the revised manuscript.

The ms reports on changing CO₂ fluxes from SE Asia since the 1980`s, diagnosed from fluxes calculated by 7 “bottom-up” land carbon models, 5 “top-down” flux inversions from atmospheric CO₂ and transport, and aboveground biomass from remote sensing. The finding that there are greater CO₂ fluxes to the atmosphere from fires during El Nino`s is not a surprise. It follows that years without El Nino`s would not be a strong source.

Response:

The key finding of this study is the strong reduction of net CO₂ emissions between the 1990s and 2000s due to the effect of CO₂ fertilization and absence of intense El Niño events during the 2000s. We tried to first point out that the CO₂ uptake in the 2000s was large enough to nearly cancel the CO₂ emissions from LUC during the same time period. Carbon fluxes from deforestation and other LUC activities still comprise a large portion of the total flux in Southeast Asia. However, our analysis emphasizes that a natural climate cycle has the potential to offset the net CO₂ emissions from deforestation and other land use change activities at the decadal level.

The finding that temperature anomalies during El Nino years, rather than precipitation anomalies or insolation anomalies, dominate the CO₂ flux anomalies (and GPP anomalies – cf Figure S11) seems counter-intuitive, and brings into question the bottom-up models.

SE Asia is typically in drought during El Niño's. Also Nemani et al. (Science 2003) shows that insolation is the dominant control on NPP in the tropics. The authors report the finding, but do not provide a mechanism for such a finding.

Response:

It is well established that increases in temperature during El Niño events reduce plant productivity in tropical regions (e.g., Jones et al., 2001; Clark et al., 2004; Hashimoto et al., 2004; Wang et al., 2013). The tight coupling between the CO₂ growth rate and tropical temperature (Wang et al., 2013; Wang et al., 2014) is probably the strongest evidence illustrating the temperature dependence of the tropical carbon cycle.

Precipitation does play an important role during El Niño, particularly as it relates to fire and its associated carbon emissions (as discussed in the original manuscript L169-178). However, there is high uncertainty as to whether precipitation can dominate the variability in CO₂ fluxes from plants in Southeast Asia. Indeed, some studies report that the variability in precipitation, most notably drought events, affects CO₂ fluxes from tropical ecosystems (e.g., Tan et al., 2013; Olchev et al., 2015). But other studies also indicate that CO₂ fluxes in tropical regions can remain unchanged during droughts because deep rooting allows plants to avoid drought stress (Goulden et al., 2004; Ichii et al., 2007) and plants adjust carbon allocation to tolerate droughts without increasing mortality (Doughty et al., 2015). The diversity of the (flux) response to precipitation in tropical ecosystems does not suggest compelling evidence of a strong, uni-directional, control of plant CO₂ exchange from precipitation in Southeast Asia.

Nemani et al. (2003) arrived at their conclusion that insolation was the primary control of NPP in the tropics based on an oversimplified empirical model of NPP, in which plant growth had predefined climate limits (see supplemental information of Nemani et al.

2003, but briefly reviewed as follows). In their model, they defined the temperature limit to NPP as monthly minimum temperatures between -5°C and 5°C , scaled between 1 and 0, wherein for temperatures above 5°C the index value was 0 and there would be, by definition, no limitation. As a consequence, temperature was precluded from limiting NPP in tropical and sub-tropical regions. Similar assumptions on climate limits (predefined ranges) were applied to precipitation and insolation, but they did not provide any clear justifications (only assumptions) for their selection of the limits. This is not a strict criticism of their work, as we know much more now than we did then.

The new version of our manuscript attempts to explain the mechanism behind the relationship between temperature anomalies and plant CO_2 exchange (L176–191 in the revised manuscript). Specifically, we demonstrate that anomalous temperature rises during El Niño events reduces the photosynthetic capacity of plants (Gross Primary Production: GPP) using the TRENDY model simulations. We also provide observational evidence from empirical upscaling of eddy flux data (FLUXCOM, Jung et al., 2017), which supports the model-based finding that the sensitivity of plant CO_2 exchange in Southeast Asia is temperature dependent.

Major concerns:

(1) SE Asia is a particularly challenging location for study, as there are few observations to constrain “bottom-up” estimates or “top-down” inversions. Consensus among the models does not mean that the models are correct. I would have appreciated a critical evaluation of the different constraints on the fluxes derived for this region, so that I could have some confidence that the modeling results something have some relevance to the real world.

Response:

We very much agree that there are only a few flux observations in tropical regions, and that any study will suffer from a limited observational dataset. To address this problem, we have taken a multiple approaches (“bottom-up”, “top-down”, and “remote sensing”), to build robustness and confidence in our findings which are now based on the agreement from three independent lines of evidence.

All three approaches for net CO₂ flux estimation depend on a different set of forcing: historical atmospheric CO₂, climate, and LUC for biosphere models; atmospheric CO₂ measurements and wind transport for atmospheric CO₂ inversions; and Vegetation Optical Depth (VOD) by microwave sensors for Δbiomass (see Methods in the revised manuscript). The agreement between these independent estimates of regional net CO₂ fluxes should not be taken for granted, especially for tropical regions where agreement between “bottom-up” and “top-down” approaches have not been addressed before this study.

(2) I am totally perplexed by the comparison with aboveground biomass carbon (ABC) data with the bottom-up and top-down CO₂ fluxes. ABC is an inventory, cross-checked against lidar-based estimates of biomass. I would have thought that ABC would be a metric for cross-checking how the various bottom-up models convert the given landuse areas into ABC. Yet ABC is interpreted as a flux in this ms. I can imagine a drought that diminishes photosynthesis without altering the biomass or killing trees. So the correlations between, and comparisons of, areally-averaged fluxes and areally-averaged ABC may be conflating forested and deforested areas.

Response:

We compared “annual change (a difference between current and preceding years)” in biomass carbon with annual CO₂ budgets by the TRENDY models and atmospheric CO₂ inversions. The annual CO₂ budget is a sum of all annual CO₂ fluxes from the terrestrial biosphere, and it is roughly equivalent to the change in annual carbon stocks (i.e., carbon in live vegetation, litter, and soil). Annual biomass change (Δ biomass in the manuscript) is not equivalent to annual CO₂ budget because it does not account for the annual changes in litter and soil organic carbon. However, it is still a useful indicator of temporal variations in net CO₂ flux because the aboveground carbon dominates the annual change in carbon stocks.

(3) The criterion for selecting land carbon models appears to be the availability of NBP output, rather than performance of the models against some metric. IPCC AR5 WG1 Figure 6.17 shows a large range in the sensitivities of (globally averaged) land CO₂ fluxes to temperature and precipitation among the 10 models that participated in the emissions-forced CMIP5 experiments. Figure 6.17 (IPCC) does not inspire confidence in land carbon models and their projections of changes. Nor does Figure S4 of this ms: LPJ-GUESS has a very heterogeneous response even to CO₂ fertilization – this demands explanation! I find Figures S9-S14 very confusing. See comment above re Figure S11 (GPP). It would have been more instructive to show for each model sensitivities of GPP and RE, rather than NBP, to the climate variables.

Response:

It is true that the selection of biosphere models for the analysis is based on the availability of NBP outputs given that modeling groups are not required to published full output, but it is at a voluntarily basis. This selection turned out to be more appropriate than

selecting the models based on performance because it is bias free; that is, we did not deliberately select the models for consistency. As the reviewer mentioned, LPJ-GUESS does have heterogeneous spatial patterns of CO₂ fluxes (Figure S4), but this is because the model simulates so-called ‘forest gap’ dynamics that involve stochastic processes of mortality. The spatial heterogeneity of CO₂ fluxes is a visually distracting, admittedly, but among all models there is a consistent pattern of CO₂ fertilization and climate sensitivity. We added text to the legend in Figure S4 to help clarify that the spatial heterogeneity of LPJ-GUESS fluxes result from model-specific mechanisms, but not CO₂ fertilization or climate sensitivity itself. We think it is important to include LPJ-GUESS because it is representative of the diversity of approaches used to model terrestrial dynamics. Even considering the spatial heterogeneity in CO₂ fluxes from LPJ-GUESS, the decadal shift in NBP and climate sensitivity is consistent among the biosphere models.

Climate sensitivity of global total NBP provided by the IPCC AR5 should be interpreted with caution because not all components of NBP are sensitive to climate. To accurately address climate sensitivity of NBP, only relevant components of NBP needs to be extracted. This is precisely the reason why we analyzed patterns of NBP by isolating component fluxes and their drivers (CO₂ fertilization effect, climate effect, and LUC effect). In the revised manuscript, we explicitly demonstrate consistency in NBP variability induced by climate (Supplementary Figure 6c), and climate sensitivity of plant CO₂ exchange (Figure 5a), GPP, and RE (Supplementary Figure 8), for all models.

(4) “Top-down” inversions typically estimate tropical land fluxes as the residuals to balance the global budget, as there are few observations in the tropics, and the intense convection dilutes the signal. Indeed, Figure S5b shows for the 2000’s, three of the 5

models show a source, and two show a sink. The authors should provide constraints on these inversions.

Response:

Indeed, the net budgets in the 2000s are different between individual atmospheric CO₂ inversion, suggesting source or sink, but the absolute values are influenced by the lack of observations in the tropics, of course. The key of this study is the decadal change in the CO₂ budget, which is independent of the absolute magnitude of the net CO₂ budget (source or sink). As such, we take advantage of the self-consistency of the dynamics of models and inversions by analyzing the relative (change among decades) and not absolute values to determine the control (sensitivity to climate, LUC) on CO₂ fluxes.

It is true that constraints (atmospheric CO₂ measurements) for tropical regions are statistically limited compared to other regions. However, Southeast Asia has an advantage over other tropical regions because of the availability of aircraft and vessel observations, which helps inversion systems to capture signals (not residuals) from the region (e.g. Niwa et al., 2012). Constraints for each inversion system are illustrated in references in Supplementary table 2, so we do not address them in the manuscript.

Minor concerns:

(1) The ms is difficult to follow, as the crucial info about the models (e.g. what is simulation 3?) and for assessing the results are in the supplemental info. The authors should remember “Supplemental” means supplemental. The ms text should be able to stand on its own.

Response:

In the introductory paragraph, we explained three types of TRENDY simulations used in the analyses (L76-80 in the original manuscript). Simulations where biosphere models were forced with varying CO₂, climate, and historical LUC is “simulation-3”, and those forced with varying CO₂ and climate is “simulation-2” and varying CO₂ only is “simulation-1”. Details of these model setups are fully explained in Methods, which we believe that is also a part of the manuscript. In the revised manuscript, “simulation-1,-2 and -3” are changed to “TRENDY S1, S2 and S3” because these are more standard terms (Stich et al., 2015).

(2) I do not know what TRENDY-v2 is about, and do not know what is in the various bottom-up and top-down models. References for the TRENDY-v2 models would have helped.

Response:

TRENDY-v2 is the updated version of TRENDY experiment for Global Carbon Project 2015 (Le Quéré, et al 2015). Unfortunately, many publications do not distinguish different versions of TRENDY despite of different sets of models in each version. To avoid confusion, we specified the version (i.e., ver.2) of TRENDY used for the analysis in Methods (L526 in the revised manuscript), and then we referred it as TRENDY (not TRENDY-v2) throughout the manuscript.

References:

Clark, D. A. Sources or sinks? The responses of tropical forests to current and future climate and atmospheric composition. *Philos. Trans. R. Soc. Lond. B Biol. Sci.* 359, 477–491 (2004).

Doughty G. E., et al. Drought impact on forest carbon dynamics and fluxes in Amazonia. *Nature* 519, 78–82 (2015).

Goulden, M. L., et al. Diel and seasonal patterns of tropical forest CO₂ exchange. *Ecol. Appl.* 14, 42–54 (2004).

Hashimoto, H. et al. El Niño–Southern Oscillation–induced variability in terrestrial carbon cycling. *J. Geophys. Res.* 109, D23110 (2004).

Ichii, K., et al. Constraining rooting depths in tropical rainforests using satellite data and ecosystem modeling for accurate simulation of gross primary production seasonality. *Glob. Change Biol.* 13, 67–77 (2007).

Jones, C. D., Collins, M., Cox, P. M., Spall, S. A. The Carbon Cycle Response to ENSO: A Coupled Climate–Carbon Cycle Model Study. *J. Climate* 14, 4113–4129 (2001).
climate and atmospheric composition. *Philos. Trans. R. Soc. Lond. B Biol. Sci.* 359, 477–491 (2004).

Jung, M. et al. Compensatory water effects link yearly global land CO₂ sink changes to temperature. *Nature*, 541, 516–520 (2017).

Le Quéré, C. et al. Global carbon budget 2015. *Earth Syst. Sci. Data* 7, 349–396, (2015).

Nemani, R. R., et al. Climate-driven increases in global terrestrial net primary production from 1982 to 1999. *Science* 300, 1560–1563 (2003).

Niwa, Y. et al. Imposing strong constraints on tropical terrestrial CO₂ fluxes using passenger aircraft based measurements. *J. Geophys. Res.* 117, D11303 (2012).

Olchev, A., et al. Response of CO₂ and H₂O fluxes in a mountainous tropical rainforest in equatorial Indonesia to El Niño events, *Biogeosciences* 12, 6655-6667 (2015).

Sitch, S., et al. Recent trends and drivers of regional sources and sinks of carbon dioxide. *Biogeosciences*, 12, 653-679 (2015).

Tan, Z.-H., et al. High sensitivity of a tropical rainforest to water variability: Evidence from 10 years of inventory and eddy flux data. *J. Geophys. Res. Atmos.* 118, 9393–9400 (2013).

Wang, W., et al. Variations in atmospheric CO₂ growth rates coupled with tropical temperature. *Proc. Natl. Acad. Sci. USA* 110, 13061–13066 (2013).

Wang, X., et al. A two-fold increase of carbon cycle sensitivity to tropical temperature variations. *Nature* 506, 212–215 (2014).

Response to reviewer 2's comments.

Kondo et al.

“Decadal carbon balance shifts controlled by land-use change and El Niño-Southern Oscillation in Southeast Asia” (previous title “Neutral-Phased El Niño-Southern Oscillation drives Southeast Asian CO₂ uptake in the 2000s”)

Major changes are marked as red in the revised manuscript.

This is study aims at showing how ENSO, or the neutral phased ENSO, influences atmospheric CO₂ fluxes in Southeast Asia. The main conclusion is that the 2000s period, when there were no strong El Ninos (termed as “neutral-phased ENSO” by the authors), saw a weakened CO₂ source compared to the 80s and 90s as well as in the most recent years. The method is comprehensive and I believe this study would make a useful contribution to the field. However, the paper would benefit from a careful revision to improve its clarity and conviction. To start with, the readers would be confused by the term “neutral-phased ENSO” to refer to the absence of strong El Nino events, rather than the actual neutral events (i.e., neither El Nino nor La Nina events).

Response:

We agree that the term “Neutral-phased ENSO” is rather ambiguous and the revised manuscript avoids this terminology (correspondingly, we changed the title).

Secondly, as far as ENSO is concerned, it is odd that there are no discussions on the role of La Nina in CO₂ uptake. For instance, one would wonder how the 2000s period “net sink” is due to the prevalent La Nina events. This issue is not clear because the authors assigned weak/moderate El Nino events, together with La Nina and neutral years, as “non

El Nino years”. There are also a few places where clarity could be improved as outlined below.

Response:

We addressed the role of La Niña thoroughly in the discussion of the revised manuscript (L235–252 and Fig. 6), including the variability in NBP during ‘neutral’ events (i.e., weak El Niño and La Niña).

First of all, the paper lacks references in ENSO and its relationship with terrestrial CO2 cycle. The authors could start with a recent paper by Kim et al. (2016, J. Climate, <http://dx.doi.org/10.1175/JCLI-D-14-00672.1>), and earlier studies such as: Hashimoto, H., and coauthors, 2004: El Niño–Southern Oscillation-induced variability in terrestrial carbon cycling. J. Geophys. Res., 109, D23110, doi:10.1029/2004JD004959. Jones, C. D., M. Collins, P. M. Cox, and S. A. Spall, 2001: The carbon cycle response to ENSO: A coupled climate–carbon cycle model study. J. Climate, 14, 4113–4129. Rayner, P. J., R. M. Law, and R. Dargaville, 1999: The relationship between tropical CO2 fluxes and the El Niño–Southern Oscillation. Geophys. Res. Lett., 26, 493–496, doi:10.1029/1999GL900008. These references can be used in illustrating the different roles between El Nino and La Nina.

Response:

With the literatures suggested, we extended discussions about the climate sensitivity of carbon fluxes (L176–214) and ENSO-carbon cycle relationship (L235–252) in the revised manuscript.

With regards to defining ENSO years, then weak/moderate El Nino events should be identified, as well as La Nina events. The analysis will be neater this way, as it will clearly distinguished the effect of El Nino events in general, and importantly providing a proper understanding of the role of ENSO (i.e., La Nina vs El Nino). In the current manuscript, the term “neutral-phased ENSO” to refer to the absence of strong El Nino is confusing, as this term is expected to refer to neutral years. Using the MEI alone is sufficient, i.e., there is no need to add an additional constraint in SOI. Please add a paper in Methods for general definition of ENSO years. This more refined classification should clarify the rest of the analysis as it should make the difference between El Nino and non El Nino years more significant.

Response:

In the revision, we first classified types of El Niño and La Niña by a rule that seasonal MEI (note that SOI is no longer used in the revised manuscript) falls within the predefined range at least one season of year, such that MEI = -2 to -1 (strong La Niña), -1 to 0 (weak/moderate La Niña), 0 to 1 (weak El Niño), 1 to 2 (moderate El Niño) and 2 to 3 (strong El Niño). To characterize the intensity of El Niño events, however, the magnitude and the duration of seasonal MEI needed to be considered. Therefore, we then defined “Intense El Niño years” as years during which the seasonal MEI values fall within MEI greater than one (moderate or strong El Niño) for at least two seasons. We chose our own method for defining ENSO conditions because, to the best of our knowledge, there is no standard method agreed upon within the scientific community (definitions vary by national institutes and depend on indices that they themselves use).

For instance the statement in L43: “In the 2000s,...resulting in a weaker CO2 loss from the land (Fig. 2c)” - is puzzling because Fig. 2c shows no values at all for the El Nino years, i.e., there is nothing to be compared with the 1980s and 1990s.

Response:

We believe that the statement still stands because the “absence” of El Niño years (intense El Niño years in the revised manuscript) is one of the causes for neutral carbon balance in the 2000s. Strong emissions in response to moderate/strong El Niño in the 1980s and 1990s did not occur in the 2000s.

This more refined classification will also enhance the difference between El Nino years (strong and weak El Ninos) and non El Nino years (i.e., La Nina and neutral events) within each decade. A supplementary analysis can be added to show the analysis for neutral events only. This will reveal the role of La Ninas. Once all these are clarified, it would be easier to understand what’s going in the 2010-2015 period when the models provide incomplete data.

Response:

Based on the new definitions of El Niño and La Niña, we addressed the role of La Niña thoroughly in the discussion of the revised manuscript (L235–252 and Fig. 6), including the variability in NBP during neutral events (i.e., weak El Niño and La Niña).

Statistical significance should be noted in Fig. 2, in comparing El Nino years in 2000s versus El Nino years in 1980s and 1990s; as well as in comparing El Nino years vs non El Nino years in each decade.

Response:

We believe that it is not technically feasible to address statistical significance on annual CO₂ budgets with limited statistics (seven biosphere models, five CO₂ inversions, and one estimate for annual biomass change). The best we can do is to show variations among estimates by different approaches.

It is not clear how the 2010-2015 NBP anomalies are obtained, given the absence of full data for that period. In Methods (L547-555), it is stated that the three-month lag correlation for the period 2010-2012 is used to project for the remaining years (2013-2015). But 3 years of data (2010-2012) is not sufficient to represent ENSO. One would need at the very least 10 years to sample ENSO events. Many researchers regard that even 10 years are not long enough. I would also encourage the authors to use available observational data for this recent period rather than just empirical derivation from the models.

Response:

Our choice of wording might have not been the best. We projected the NBP-MEI relationship to the period 2013–2015 (updated to the period 2013–2016 in the revised manuscript) by using the TRENDY and inversion data for the period 2010–2012 (as 2012 is the end of simulation period of the TRENDY) and then for the period 2013–2016 by the relationship obtained for the period 1980-2009 (not 2010–2012). We chose the empirical projection based on the models because, in that way, the data would be self-consistent throughout the study period. Recent observational NBP data that cover the period 2010-2016 may be available in near future, but they are still yet to come.

L86-88: In contrast to what's stated here, the variations in simulation-2 appear correlated with the atmospheric CO₂ inversions and annual biomass change (Fig. 1a), despite the offset in the mean which should be due to LUC. The corresponding correlations for simulation-2 should also be shown in Fig. 1a to support the statement.

Response:

The simulation-2 (TRENDY S2 in the revised manuscript) accounts for variability in atmospheric CO₂ and climate, and therefore, some degree of correlation is expected when comparing the TRENDY modeled fluxes to inversion data and the Δ biomass. However, a comparison between the data from simulation-3 (TRENDY S3 in the revised manuscript), which includes LUC simulation, and inversion or biomass data should show higher correlations. We made this point explicit in the figure below (Fig. S1).

Figure S1. Relationships between ensemble averages of annual NBP from TRENDY S3 and atmospheric CO₂ inversions (top-left), annual biomass change (top-right), and between that from TRENDY S2 and atmospheric CO₂ inversions (bottom-left), annual biomass change (bottom-right).

L99 (and elsewhere): “towards a net sink from the 1990s to the 2000s” shouldn’t this be “towards a weaker net source from the 1990s to 2000s” as we are looking at the orange bar (simulation-3) which still shows a weak positive value. In fact this should be clearer in Fig. 1d where the net value could also be added as verification.

Response:

We avoided to use the phrase “...towards a net sink” in the revised manuscript.

The sentence in L88-92 is not a good read. What did you mean by “LUC variability (...) changed patterns of the spatial variability”? “..., flux changed” what flux?

Response:

The sentence is revised in the new manuscript (L95–108).

Figure 1, caption: “Interannaul”

Response:

Thank you, this typo is now corrected in the new manuscript.

Figure 2: Shouldn’t there be an error bar on each of the orange triangles?

Response:

In this analysis, error bars represent variations among model estimates. Therefore, we did not show an error bar for fire emissions by CLM (the orange triangle).

L146-158: it should be stated if the differences described here are statistically significant.

Response:

As we mentioned before, it is not technically feasible to address statistical significance on annual CO₂ budgets with limited statistics.

L85: “resulting in” should be “as indicated by”

Response:

The sentence is revised as advised.

L541: “forwards” should be “leads”

Response:

The word is revised as advised.

L543-544: “with” should be “at”

Response:

The word is revised as advised.

Supp. Table 4 caption: indicate the period of analysis (1980-2009?)

Response:

The caption is revised as advised.

Please indicate statistical significance in Supp. Fig. 6.

Response:

We explained this in the above.

“Neutral-phased ENSO in the 2000s” sounds quite odd. There are certainly ENSO events in the 2000s, such as the 2002/03, 2006/07 El Nino and 2005/06, 2007/08, and 2008/09 La Nina events.

Response:

The term “neutral-phased ENSO” is no longer used in the revised manuscript.

Response to reviewer 3's comments.

Kondo et al.

“Decadal carbon balance shifts controlled by land-use change and El Niño-Southern Oscillation in Southeast Asia” (previous title “Neutral-Phased El Niño-Southern Oscillation drives Southeast Asian CO₂ uptake in the 2000s”)

Major changes are marked as red in the revised manuscript.

Reviewer #3 (Remarks to the Author):

In the paper “Neutral-Phased El Niño-Southern Oscillation drives Southeast Asian CO₂ uptake in the 2000s”, authors used multiple modelling approaches to estimate CO₂ fluxes and reached the conclusion that “CO₂ release associated with El Niño-induced anomalies were insignificant during the 2000s”. Author also suggested that “Strong El Niño events in 2010–2015 indicate that neutral-phased ENSO lasted only during the 2000s and in the future might lead to increased vulnerability of carbon stocks in Southeast Asia.”

Authors do not quantify and separate the impact of LUC on CO₂ flux in the past decades from other factors such as ENSO. Therefore, there are not enough evidences to support authors statements, such as “the moderate climate in the 2000s represents a unique case where a natural climate cycle acted to mitigate CO₂ emissions in Southeast Asia”. Reviewer would recommend further analysis to support the final conclusions and findings of this manuscript.

Response:

Contrary to criticism by the reviewer, the result of our analysis is largely based on isolated attributions to net CO₂ flux (i.e., CO₂ fertilization, climate, and LUC effects) for each of the past three decades. So we do not simply understand the reason for the comment

“Authors do not quantify and separate the impact of LUC on CO₂ flux in the past decades from other factors such as ENSO”.

Using currently available model and observational data from different approaches, we explicitly demonstrated that LUC emissions in the 2000s are comparable to that in the 1990s, in addition that a net source in the 1990s was largely reduced to carbon neutral in the 2000s because of the effect of CO₂ fertilization and the absence of intense El Niño events. We believe that our analysis very much addresses the concern of this reviewer.

Reviewers' comments:

Reviewer #1 (Remarks to the Author):

This ms is a contribution to the large body of literature on variability of atmospheric growth rate, tropical ecosystems and ENSO. The authors combine satellite observations of biomass change, bottom simulations and top-down inversions to determine NBP. Unlike many other studies, it is focused on SE Asia, and includes periods when El Ninos are not intense. The authors find that during the decade of the 2000's when intense El Ninos were absent, the tropical ecosystems of SE Asia had greater CO₂ uptake (NBP), mainly because of the absence of fires. This is consistent with published finding of greater CO₂ release during strong El Ninos.

I strongly support the methodology of the investigation, which uses multiple approaches and observations to constrain the inversion results. I am disappointed however by the discussion. One comes away with the obvious conclusion – absence of fire emissions associated with strong El Ninos yields a reduced net source or stronger net sink. The ms would be much stronger if it gives insight into the processes (not just parameters) that control CO₂ exchanges in the varying climate.

LUC used here is a jargon term or a dataset for computer simulations, and does not give the reader any clue about the processes/practices involved. The authors write the accelerated rate of conversion of forest to palm plantations since the 1980's. Is this the dominant practice? L170: "the magnitude of LUC emissions was nearly unchanged regardless of El Nino conditions". It seems like LUC is a necessary factor throughout the period, and may not contribute to decadal changes. This interpretation would contradict the title of the ms.

The section (L175) on climate sensitivity OF (not TO) carbon fluxes is very weak, and comprises mainly speculations and caveats. In the 1980s and 1990s, the models indicated CO₂ uptake by plants (separate from fire emissions) was a strong source during intense El Nino year, as that photosynthesis is reduced during periods of high temperatures. In the 2000's without intense El Nino's, NBP still indicates a CO₂ source to the atmosphere, albeit a reduced source. Why? The discussion about insolation and precipitation is confusing, as high temperatures in SE Asia are typically accompanied by low precipitation, low cloud cover, and high insolation anomalies. It seems premature to dismiss these coupled factors.

Reviewer #2 (Remarks to the Author):

The paper has been significantly improved. However, the classification of ENSO events is still awkward. The authors classify events with MEI between 0 to 1 as "weak El Nino events" and MEI between -1 and 0 as "weak/moderate La Nina events". But this would include non ENSO years as well which would mistakenly be considered as weak ENSO events. This can be easily corrected (e.g., weak/moderate ENSO should be MEI within 0.5-1, and MEI between 0 to 0.5 can be considered as neutral events).

There are typos:

E.g., L224: "REgional"

L252: "lessor"

L256: "2014/15 El Nino" or "2015/16 El Nino" ? 2014/2015 was not an El Nino event. The tropical Pacific showed some warming but that wasn't a proper El Nino event. 2015/2016 was a big El Nino.

L464: "triable"

Response to reviewer 1's comments.

Kondo et al.

“Decadal carbon balance shifts controlled by land-use change and El Niño-Southern Oscillation in Southeast Asia”

Major changes are marked as red in the revised manuscript.

This ms is a contribution to the large body of literature on variability of atmospheric growth rate, tropical ecosystems and ENSO. The authors combine satellite observations of biomass change, bottom simulations and top-down inversions to determine NBP. Unlike many other studies, it is focused on SE Asia, and includes periods when El Ninos are not intense. The authors find that during the decade of the 2000's when intense El Ninos were absent, the tropical ecosystems of SE Asia had greater CO₂ uptake (NBP), mainly because of the absence of fires. This is consistent with published finding of greater CO₂ release during strong El Ninos. I strongly support the methodology of the investigation, which uses multiple approaches and observations to constrain the inversion results. I am disappointed however by the discussion. One comes away with the obvious conclusion – absence of fire emissions associated with strong El Ninos yields a reduced net source or stronger net sink. The ms would be much stronger if it gives insight into the processes (not just parameters) that control CO₂ exchanges in the varying climate.

Response:

The conclusion of the analysis is that the increased CO₂ uptake in the 2000s due to the enhanced CO₂ fertilization and absence of strong El Nino events is large so much as to reduce a substantial proportion of CO₂ release from ongoing land use changes. The reduction of a net source by the milder climate in the 2000s may be evident for those who know Southeast Asia, but we emphasize impact of its effect (the absence of fire is only a

part of it), which acts as a major control of the decadal net CO₂ flux between the 1990s and 2000s.

Revision added more analyses on climate sensitivity of carbon fluxes (L175-203 in the revised manuscript). Specifically, we added Standardized Precipitation Index (SPI) to the analysis in addition to temperature, precipitation, and short-wave radiation (see Methods section of in the revised manuscript: L663-674). Because it accounts for a cumulative effect of precipitation, we considered that SPI may be a better indicator of water availability for plants than precipitation. As a result, we found that CO₂ uptake by plants decreased due to reduced GPP during periods with large increases in temperature and decreases in water availability (indicated by SPI) associated with El Niño. This result is consistently found not only in TRENDY models, but also in the empirically up-scaled eddy flux data (FLUXCOM).

LUC used here is a jargon term or a dataset for computer simulations, and does not give the reader any clue about the processes/practices involved. The authors write the accelerated rate of conversion of forest to palm plantations since the 1980's. Is this the dominant practice? L170: "the magnitude of LUC emissions was nearly unchanged regardless of El Nino conditions". It seems like LUC is a necessary factor throughout the period, and may not contribute to decadal changes. This interpretation would contradict the title of the ms.

Response:

The LUC dataset used in this study lacks details about the processes and practices, but that is because the information contained in the dataset are limited by the FAO statistics (annual changes of cropland and pastureland area, and wood harvest). Although

information about species chosen for plantations in Southeast Asia (such as oil palm and rubber trees) are not provided, the dataset still distinguishes secondary forests from primary forests (see Supplementary Information Fig. 13). The information about plantations are included as secondary forests in the dataset.

LUC emissions contribute to the decadal variability of net CO₂ flux as we showed that the shift towards a net source from the 1980s to the 1990s (Fig. 1c) is due to the increased LUC emissions in the 1990s (see Fig. 1d). Therefore, the result does not contradict to the title of the manuscript.

The section (L175) on climate sensitivity OF (not TO) carbon fluxes is very weak, and comprises mainly speculations and caveats. In the 1980s and 1990s, the models indicated CO₂ uptake by plants (separate from fire emissions) was a strong source during intense El Nino year, as that photosynthesis is reduced during periods of high temperatures. In the 2000's without intense El Nino's, NBP still indicates a CO₂ source to the atmosphere, albeit a reduced source. Why?

Response:

CO₂ uptake by plants increased due to the enhanced CO₂ fertilization and absence of strong El Nino events in the 2000s, but LUC emissions in the 2000s are still large with the magnitude comparable to that in the 1990s (Fig. 1d). Therefore, NBP is a weak net source (or neutral if we consider the range of uncertainty) in the 2000s despite a reduced source from the 1990s (Fig. 1c).

The discussion about insolation and precipitation is confusing, as high temperatures in SE Asia are typically accompanied by low precipitation, low cloud cover, and high insolation anomalies. It seems premature to dismiss these coupled factors.

Response:

It is not necessarily true that high temperatures in Southeast Asia accompany low precipitation, low cloud cover, and high insolation anomalies (see Supplementary Figure 7). As we addressed in the manuscript, plant CO₂ exchange is not sensitive to precipitation, but it is to SPI. Please see the revised section “Climate sensitivity of carbon fluxes in Southeast Asia” (L175-203). We believe that cumulative precipitation over preceding months is more effective to plant CO₂ exchange than simple monthly precipitation.

Response to reviewer 2's comments.

Kondo et al.

“Decadal carbon balance shifts controlled by land-use change and El Niño-Southern Oscillation in Southeast Asia”

Major changes are marked as red in the revised manuscript.

The paper has been significantly improved. However, the classification of ENSO events is still awkward. The authors classify events with MEI between 0 to 1 as “weak El Nino events” and MEI between -1 and 0 as “weak/moderate La Nina events”. But this would include non ENSO years as well which would mistakenly be considered as weak ENSO events. This can be easily corrected (e.g., weak/moderate ENSO should be MEI within 0.5-1, and MEI between 0 to 0.5 can be considered as neutral events).

Response:

Thank you for the suggestion. Maybe it is better not to distinguish weak ENSO events from non ENSO events because of unclear signals from those events. We discarded weak El Nino/La Nina from the classification, and instead defined “weak ENSO” for MEI values within -1 and 1. The rest of ENSO events are classified as follows: moderate El Nino for MEI values between 1 and 2, and strong El Nino for MEI values greater than 2, moderate and strong La Nina for MEI values less than -1 (Fig. 6).

There are typos:

e.g., L224: “REgional”

L252: “lessor”

*L256: “2014/15 El Nino” or “2015/16 El Nino” ? 2014/2015 was not an El Nino event.
The tropical Pacific showed some warming but that wasn't a proper El Nino event.
2015/2016 was a big El Nino.*

L464: “triable”

Response:

Thank you, these typos are now corrected in the revised manuscript. Only “REgional” is correctly capitalized as the formal abbreviation for RECCAP is as follows, REgional Carbon Cycle Assessment and Processes (RECCAP).

Reviewers' Comments:

Reviewer #1:

Remarks to the Author:

The revisions have improved the readability of the manuscript, and the message is much clearer to this reader. I have one remaining technical question. Suppl Figure 7 shows 3-month precipitation anomalies and SPI's. I do not understand why the 3-month SPIs (derived via a gamma function fitted to 3-month cumulative precipitation) do not follow 3-month averaged precipitation anomalies: for example, it is difficult to trace the the dips in SPI around the dark and light grey bars circa 1982/1983 and 1997/1998 back to precipitate anomalies. This is important point to clear up as a conclusion of the study is that moisture availability influences CO2 flux.

Response to reviewer 1's comments.

Kondo et al.

“Carbon balance shifts controlled by land use change and El Niño-Southern Oscillation in Southeast Asia”

Major changes are marked as red in the revised manuscript.

The revisions have improved the readability of the manuscript, and the message is much clearer to this reader. I have one remaining technical question. Suppl Figure 7 shows 3-month precipitation anomalies and SPI's. I do not understand why the 3-month SPIs (derived via a gamma function fitted to 3-month cumulative precipitation) do not follow 3-month averaged precipitation anomalies: for example, it is difficult to trace the the dips in SPI around the dark and light grey bars circa 1982/1983 and 1997/1998 back to precipitate anomalies. This is important point to clear up as a conclusion of the study is that moisture availability influences CO2 flux.

Response:

As described in the Methods section, calculation of SPI is based on cumulative precipitation data for *a moving window* of different length of months, which differ in temporal variability from cumulative precipitation for *a fixed window*. This is a reason why, even with a window of the same length, three-month precipitation anomalies and SPI 3-month do not follow a similar pattern in Supplementary Figure 7 of the previous manuscript.

Changes from the previous submission.

All changes are marked as red in the revised manuscript.

In accordance with the suggested changes from the editor and journal format requirements, we have modified the manuscript as follows,

1. Changed the title from “*Decadal carbon balance shifts controlled by land-use change and El Niño-Southern Oscillation in Southeast Asia*” to “*Carbon balance shifts controlled by land use change and El Niño-Southern Oscillation in Southeast Asia*” in order to keep the number of words within the limit.
2. Removed abbreviations (i.e. LUC) from the abstract, speech marks (‘’), and numbered or bulleted lists from the manuscript.
3. Revised the final paragraph of the Introduction such that it briefly summarizes the major results of the analysis.
4. Removed abbreviations from sub-headings.
5. Modified sub-headings to fit within 60 characters.
6. Moved the personal statement at the end of the main text to the Acknowledgements.
7. Replaced a rainbow color bar with a blue-orange color bar for all spatial maps in the main and Supplementary figures.
8. Added a figure title to the figure 5.
9. Added the definition of error bar in the figure captions.
10. Modified the manuscript such that all supplementary figures and tables are cited in the main text in order.